# Air quality impacts of stratospheric aerosol injections are likely small and mainly driven by changes in climate, not aerosol settling

Cindy Wang[1], Daniele Visioni[1], Glen Chua[2,3], and Ewa M. Bednarz[4,5]

[1]Department of Earth and Atmospheric Sciences, Cornell University, Ithaca, NY, USA
[2]NASA Goddard Institute for Space Studies, NY, NY, USA
[3]Oak Ridge Associated Universities, NASA NPP Program, Oak Ridge, TN, USA
[4]Cooperative Institute for Research in Environmental Sciences (CIRES), University of Colorado Boulder, Boulder, CO, USA
[5]NOAA Chemical Sciences Laboratory (NOAA CSL), Boulder, CO, USA

**Correspondence:** Cindy Wang (cindywang@cornell.edu)

**Abstract.** Stratospheric aerosol injection (SAI) is a proposed climate intervention method to offset future global warming through increased solar reflection in the stratosphere, but its broader environmental and public health implications are yet to be thoroughly explored. We use three large ensembles of fully coupled CESM2-WACCM6 simulations to assess changes in mortality attributable to fine particulate matter ($PM_{2.5}$) and surface ozone exposure ($O_3$). Maintaining temperatures at 1.5°C

above preindustrial levels through SAI is projected to yield a modest 0.4% (ensemble range: -1.9% to +1.5%) reduction in pollution-related mortality relative to middle-of-the-road climate change scenario, reflecting a 1.3% (-2.3% to -0.6%) reduction in ozone-related deaths and a 0.9% (-0.4% to +2.1%) increase in $PM_{2.5}$-related deaths. The spread among ensemble members underscores the influence of internal variability and highlights the importance of ensemble-based analyses when assessing the potential health impacts of climate intervention strategies. We find that global $PM_{2.5}$ mortality changes exhibit little

sensitivity to injected sulfate amounts, with the most variability driven by precipitation-mediated changes in non-sulfate $PM_{2.5}$ species (e.g., dust and secondary organic aerosols), whereas ozone-related mortality is primarily driven by surface cooling and hemispheric asymmetries in stratospheric-tropospheric exchange and ozone transport. However, our results heavily reflect the specific forcing patterns of the SAI scenarios used; our estimates are also limited by model shortcomings, including omitting the effects of aerosols in the photolysis scheme - which might limit UV-driven changes and impact surface ozone rates - or

not including nitrate aerosols. Within our framework, we find that SAI impacts on pollution-related mortality are modest but regionally heterogeneous, and that the magnitude of the SAI-driven changes is smaller than the improvements expected from near-term air quality policies planned or implemented within the same time frame.

## 1 Introduction

Stratospheric aerosol injection (SAI) is a proposed climate intervention strategy that could help ameliorate the effects of

anthropogenic global warming. It involves the release of sulfur dioxide ($SO_2$), which serves as a precursor to sulfate aerosols, into the stratosphere, in order to increase Earth's albedo and lower surface temperatures. This approach draws on the observed cooling effects of large explosive volcanic eruptions (McCormick et al., 1995; Robock, 2000) and has been shown in climate

model simulations to reduce global mean surface temperatures relative to scenarios without such intervention (Tilmes et al., 2018; Kravitz et al., 2015). However, despite its potential to offset some of the warming caused by greenhouse gases, SAI raises numerous questions about its broader environmental, societal, and health-related consequences. One key concern is the impact of SAI on public health and air quality (Tracy et al., 2022). In terms of air quality, the main drivers of changes would include the direct impacts of sulfate particles on surface fine particulate matter ($PM_{2.5}$), and changes in surface ozone exposure ($O_3$); the latter would be a function of changes in stratosphere-to-troposphere $O_3$ transport and in-situ changes in tropospheric ozone chemistry driven by the SAI-induced changes in temperatures and photolysis.

This study aims to assess the effects of SAI on air pollution mortality, particularly through changes in surface $PM_{2.5}$ and surface ozone ($O_3$), by using a fully-coupled modeling approach with the Community Earth System Model (CESM2) Whole Atmosphere Climate-Chemistry Model (WACCM6), which includes interactive aerosols and detailed representations of stratospheric and tropospheric chemistry. While using non-coupled model approaches allow one to better separate and quantify the contribution of single factors, a fully-coupled model allows for the simulation to include the interaction between aerosols, atmospheric composition and climate: what might be lost in precision in the diagnosis of changes can be gained in providing a more holistic picture of the overall expected change.

Previous studies have looked into the health impacts of SAI due to air quality changes (Eastham et al., 2018; Visioni et al., 2020; Moch et al., 2023; Harding et al., 2024). These efforts have either relied on more idealized modeling frameworks and/or simplified mortality estimation methods. In particular, both Eastham et al. (2018) and Moch et al. (2023) used chemical transport models (CTMs) to quantify global mortality effects from SAI, including contributions from changes in air quality and UV-B exposure. While CTMs like GEOS-Chem have been widely applied to study air-pollution-related health outcomes (Norman et al., 2025), they are fundamentally limited in capturing the dynamical and chemical feedbacks relevant to SAI. For example, in Eastham et al. (2018), the aerosol size distribution was prescribed offline assuming a fixed lognormal distribution centered at 0.16 $\mu$m. The use of a CTM also precludes accounting for interactive changes in stratosphere–troposphere exchange (STE), temperature-dependent tropospheric chemistry, and large-scale circulation responses to SAI. As a result, such models tend to predict spatially uniform decreases in stratospheric ozone and, consequently, reductions in tropospheric ozone via STE, without accounting for compensating changes in transport or chemistry.

Harding et al. (2024) further used similar estimates as Eastham et al. (2018) and compared them against estimates of SAI impact on temperature-attributable mortality in the GFDL/FLOR model, in which the radiative forcing from geoengineering was simulated by reducing the solar constant. While solar dimming provides a simplified means of approximating the cooling effects of geoengineering, such approaches would not account for the spectrally dependent scattering and absorption properties of stratospheric aerosols, nor would it adequately capture the associated chemical and dynamical feedbacks, particularly those influencing ozone and STE (Visioni et al., 2021; Bednarz et al., 2022).

Finally, Xia et al. (2017) examined the impacts of SAI on tropospheric ozone through the use of a low-top version of CESM2, simulating SAI itself through prescribing an aerosol distribution (therefore with no changes in stratospheric aerosols settling and deposition) or through a solar constant reduction; they found that surface ozone generally decreases as a consequence of

SAI, with some significant differences between solar dimming and SAI driven by changes in stratospheric ozone and STE, but did not quantify the resulting health implications of changes in surface ozone on human exposure.

In this study, we use simulations from the Assessing Responses and Impacts of Solar intervention on the Earth system with Stratospheric Aerosol Injection (ARISE-SAI) experiment using CESM2-WACCM6 (Davis et al., 2023; Gettelman et al., 2019), which simulates SAI with injections at four discrete latitudinal points (15°S, 15°N, 30°S and 30°N) to maintain global mean surface temperatures at the 1.5°C (ARISE-SAI-1.5) or 1.0°C (ARISE-SAI-1.0) above preindustrial levels (Richter et al., 2022). This model includes interactive aerosol processes, whose evolution is simulated through the use of a modal approach (Liu et al., 2016), and a detailed representations of tropospheric and stratospheric chemistry (Emmons et al., 2020; Tilmes et al., 2023), allowing us to assess how SAI influences air pollution and associated health risks through coupled changes in atmospheric temperatures, transport, and chemistry. Compared to previous studies, our approach provides a more realistic representation of injection strategies and chemistry-climate interactions, improving estimates of pollution-driven mortality. While this provides an important advance beyond earlier studies, some limitations remain: for example, in CESM2(WACCM6) photolysis rates are calculated using lookup tables, taking into account the overhead ozone column and clouds but excluding the effects of aerosols, thereby reducing the effect SAI aerosols could have on tropospheric photochemistry and ozone. Our results should therefore be viewed as a further step toward understanding these interactions, with important knowledge gaps that future studies will need to address.

Another key contribution of this study is the explicit quantification of model internal variability in estimates of air pollution and associated health impacts. Modeled air pollutant concentrations are sensitive to changes in climate and dynamics which in turn are affected by model internal variability. This could be especially important when the changes in surface air pollution arise from climate system adjustments due to SAI rather than from changes in surface emissions. While this source of uncertainty is often underexplored in the literature (e.g., it cannot be easily assessed based on CTM results), our use of a 10-member ensemble of coupled simulations allows us to highlight its substantial influence on $PM_{2.5}$ and ozone concentrations, and the associated mortality outcomes. In the following sections, we evaluate the effects of SAI on surface air quality and associated health outcomes by analyzing changes in $PM_{2.5}$ and ozone exposure, estimating attributable mortality using epidemiological risk functions, and characterizing the spatial and ensemble variability in these impacts on global and regional scales.

## 2  Methods

### 2.1  Model description

Simulations were conducted using the CESM2(WACCM6) (Gettelman et al., 2019; Davis et al., 2023), a fully coupled ocean-atmosphere model with interactive tropospheric and stratospheric chemistry and aerosols. The model simulates aerosol formation and growth through an interactive, two-moment modal aerosol microphysics scheme (MAM4; Liu et al. (2016)), allowing sulfate aerosols to evolve over time based on the simulation of nucleation, coagulation, condensation and removal processes. However, MAM4 uses assumptions of internal mixing for the size distribution of different species, whereas mass is tracked separately (Visioni et al., 2022). While stratospheric and tropospheric chemistry are fully interactive, photolysis rates

are calculated using lookup tables, taking into account the overhead ozone column and clouds but excluding the effects of aerosols (Kinnison et al., 2007), thus excluding the direct effect of the aerosols on actinic fluxes (Michelangeli et al., 1992; Palancar et al., 2013).

Simulated $PM_{2.5}$ components include sulfate ($SO_4$), secondary organic aerosols (SOA), primary organic matter (POM), salt, dust and black carbon (BC). However, the model does not include explicit ammonium or nitrate aerosol chemistry, which can

be important contributors to $PM_{2.5}$, particularly in amonia-rich regions (Nolte et al., 2018; Hancock et al., 2023). This omission may lead to an underestimate of absolute $PM_{2.5}$ concentrations and associated health impacts in certain areas, and future work should aim to understand if such an omission could impact heavily our conclusions: interactions between the formation of nitrate and sulfate aerosols are complex (Liu et al., 2020), and while recent observations have shown that it is the absence of sulfate aerosols that favors fine particulate nitrate formation in some environment (Wen et al., 2023; Wei and Tahrin, 2024),

more work is needed to understand what impact this would have under SAI scenarios. Furthermore, Hancock et al. (2023) has indicated that estimates of $PM_{2.5}$ may overestimate the contribution of dust due to the inclusion of larger dust particles.

These limitations notwithstanding, the interactive chemistry–climate framework of WACCM allows us to capture coupled meteorological, chemical, and radiative feedbacks that are central to evaluating the air quality response to stratospheric aerosol injection (Tilmes et al., 2019). CESM2(WACCM6) has been evaluated against earlier model versions and observations–

105 including NASA ATom aircraft profiles (Tilmes et al., 2019), Tropospheric Ozone Assessment Report (TOAR) surface ozone data (Emmons et al., 2020), and Measurements of Pollution in The Troposphere (MOPITT) carbon monoxide observations (Schwantes et al., 2020)–showing good agreement with ozonesonde data and seasonality of surface ozone, though with some regional spatial biases. Previous evaluations have also shown that WACCM reproduces the large-scale distributions of tropospheric ozone and key pollutants, as well as climatological patterns of aerosols, with skill comparable to other climate

models (Griffiths et al., 2021; Hancock et al., 2023). These assessments further support the suitability of this model for investigating the relative changes in air quality under SAI.

## 2.2  Simulations

The baseline ensemble (i.e. without SAI) follows the Shared Socioeconomic Pathway 2 with middle-of-the-road increases in greenhouse gas emissions, leading to a radiative forcing of 4.5 W/m$^2$ by 2100 (Fricko et al., 2017), and is hereafter referred to

115 as SSP2-4.5. In the ARISE-SAI-1.5 ensemble, under the same emission scenario, sulfur dioxide ($SO_2$) is injected annually at four fixed latitudes (15°N, 15°S, 30°N, 30°S) at approximately 21.5 km of altitude starting in year 2035, and run until 2070, with injection rates adjusted at the beginning of each year to offset continuing warming under the SSP2-4.5 emissions pathway, with the aim of maintaining global mean surface temperatures and their large-scale gradients at the 1.5°C above preindustrial level (defined as the mean over 2020-2039 to ensure better consistency with other climate models, (Visioni et al., 2024)). The

120 ARISE-SAI-1.0 simulations follow the same protocol, but SAI is used to cool by a further 0.5°C compared to the targets in ARISE-SAI-1.5.

A 10-member ensemble is produced for all three cases to account for internal climatic variability (Richter et al., 2022). For regional assessments of mortality and mortality-related factors, we will focus our analyses on the ARISE-SAI-1.5 case, whereas

results from the ARISE-SAI-1.0 will be provided for global, temporal and injection-related analyses in order to highlight the
linearity (or lack thereof) of the SAI response with the injection rates.

## 2.3 Calculation of exposure and mortality

Here we describe how we calculated the impact on mortality rates attributed to changes in the simulated changes in ambient
surface $PM_{2.5}$ and $O_3$. All mortality estimates in future scenarios are calculated using the fixed 2020 population distribution:
this approach isolates the effects of air quality changes by removing confounding influences from projected population growth
or redistribution.

Mortality is estimated using the health impact function (EPA, 2015):

$$M_{i,d,a,t} = \text{BMR}_{d,a,t} \times \text{P}_{i,a,2020} \times AF_{i,d,a,t} \tag{1}$$

Where $M_{i,t}$ is the mortality for CESM grid $i$ from disease $d$ for age group $a$ and year $t$; P is the number of population in
2020 with each age group $a$ in grid $i$; BMR is the national base mortality rate for disease $d$, age group $a$ and year $t$; $AF$ is the
attributable fraction which estimates the proportion of deaths in a population that can be attributed to a specific exposure to
disease $d$ or risk factor from epidemiological studies. For $PM_{2.5}$, we use the $AF$ associated with noncommunicable diseases and
lower respiratory infections (NCD+LRI). For ozone, we use the $AF$ associated with cardiovascular and respiratory diseases.
Whereas previous studies (Eastham et al., 2018) attributed $PM_{2.5}$ exposure to cardiovascular and respiratory diseases and
ozone exposure solely to respiratory diseases, here we attribute cardiovascular disease to ozone exposure, which aligns with
more recent epidemiological findings (Sun et al., 2024; Niu et al., 2022) and improves the completeness of ozone-related health
impact assessments.

For $PM_{2.5}$, $AF$ is calculated using the exposure–response function from the Global Exposure Mortality Model (GEMM;
Burnett et al. (2018)), which provides improved estimates across a wide range of ambient $PM_{2.5}$ concentrations. GEMM is
particularly effective in low-income and high-pollution regions where the older Integrated Exposure–Response (IER) functions
tend to underperform due to limited observational data and less robust extrapolation at high exposure levels (Burnett et al.,
2014, 2018; Burnett and Cohen, 2020):

$$AF_{i,d,a,t} = 1 - \frac{1}{RR_{i,d,a,t}}; \text{ where } RR_{i,d,a,t} = \exp^{\frac{\theta \times log(\frac{C_{i,t}}{\alpha+1})}{1+\exp^{-\frac{C_{i,t}-\mu}{v}}}} \text{ and } RR_{i,d,a,t} = 1 \text{ when } C_{i,t} < 2.4 \ \mu g/m^3 \tag{2}$$

Where C is the ambient $PM_{2.5}$ concentration ($\mu g/m^3$); $RR$ is the relative risk of morality at any concentration; $\theta$, $\alpha$, $\mu$ and
$v$ are empirical coefficients from the GEMM which are specific for each age group.

For ozone-attributable mortality, we convert surface ozone to the ozone season maximum daily 8-hour average (OSMDA8;
ppb) using hourly surface $O_3$ data for each experiment and each ensemble member. OSMDA8 calculates the highest 6-month
rolling average daily 8-hour average ozone concentration, which reflects the highest average ozone concentration over a 6-
month period. OSMDA8 is the metric used by the Global Burden of Disease (GBD) (Brauer et al., 2024) for quantifying the

health effect from long-term ozone exposure and is used in the World Heath Organization's air quality guidelines (WHO, 2021, License: CCBY-NC-SA3.0IGO). To calculate the ozone-attributable risk fraction, we calculate the $AF$ for cardiovascular and respiratory disease separately and then combine the associated mortality.

$$AF_{i,d,a,t} = 1 - \exp^{-\beta(X_{i,t} - X_{\min})}; \text{ where } AF = 0 \text{ when } X_{i,t} < X_{\min} \tag{3}$$

Where X represents the spatially and temporally resolved grid-cell level OSMDA8; $X_{\min}$ represents the theoretical minimum risk exposure concentration and $\beta$ represents a model-parameterized slope of the log-linear relationship between concentration and health from epidemiological studies. For chronic respiratory disease mortality, we apply a $\beta$ of $\ln(1.06)$ per 10 ppb ozone (95% confidence interval (CI) 1.03-1.10) derived by GBD 2019 (Jerrett et al., 2009; Malashock et al., 2022; Murray et al., 2020). For cardiovascular disease mortality, we apply a $\beta$ of $\ln(1.028)$ per 10 ppb ozone (95% CI 1.010-1.047) (Sun et al., 2024). A summary of the RR and disease $d$ used to calculate mortality associated with $PM_{2.5}$ and $O_3$ is provided in Table 1.

| Cause | Disease ($d$) | Minimum exposure concentration | Source |
|-------|---------------|-------------------------------|--------|
| $PM_{2.5}$ | Noncommunicable diseases & lower respiratory infections (NCD+LRI) | 2.4 $\mu g/m^3$ | Burnett et al. (2018) |
| Ozone | Cardiovascular diseases | 40 ppb | Sun et al. (2024) |
| | Respiratory diseases | 32.4 ppb | Malashock et al. (2022) |

**Table 1.** Summary of risk functions used for estimating attributable mortality. Minimum exposure concentrations correspond to the theoretical minimum risk exposure levels for each pollutant-health outcome pair.

Our BMRs are drawn from the International Futures (IFs) health model, providing dynamic, age and disease-specific mortality projections consistent with policy interventions following the SSP2-4.5 scenario (Hughes et al., 2014). The IFs health model is a comprehensive, integrated modeling platform used to explore long-term global health dynamics. This represents a more realistic approach compared to the use of static BMRs in previous studies (Eastham et al., 2018).

Population (P) for each age group was calculated by using the global population density dataset based on Shared Socioeconomic Pathways (SSP) (Jones and O'Neill, 2020) and the ratio of the population for each age group to the total population retrieved from the SSP database developed by the International Institute for Applied Systems Analysis and the National Center for Atmospheric Research (Riahi et al., 2017; Samir and Lutz, 2017) for each country. The raster of nation-states was retrieved from the Gridded Population of the World, Version 4 (GPWv4): National Identifier Grid (Center for International Earth Science Information Network (CIESIN) - Columbia University, 2018) and is used to aggregate the calculated mortality to country-level mortality estimates. We further categorize the world into 21 regions following the GBD Study based on epidemiological similarities and geographic proximity.

Other studies estimating air pollution-related mortality have typically calculated mortality uncertainty based on the central intervals of the parameters used in the $AF$ calculations (Peng et al., 2021; Eastham et al., 2018). However, less attention has been given to the uncertainty arising from internal model variability: this is important as internal variability can drive regional air quality differences (Fiore et al., 2015). Thus, our analysis account for uncertainty arising from climate ensemble spread, while applying central estimates for $\beta$ (for ozone) and $RR$ (for $PM_{2.5}$).

## 3 Results

In Fig. 1, and in the subsequent mortality analysis, we present changes in surface $PM_{2.5}$, ozone, temperature ($T_s$), and total precipitation in three ways: (1) the 2060–2069 average from the ARISE-SAI-1.5 simulation minus 2030-2039 average from SSP2-4.5, illustrating the change under SAI implementation; (2) the 2060–2069 average from SSP2-4.5 minus 2030-2039 average from SSP2-4.5, representing changes under the SSP2-4.5 pathway without SAI; and (3) the difference between the 2060–2069 averages of the ARISE-SAI-1.5 simulation and SSP2-4.5, showing the direct impact of SAI by comparing a future with SAI to one without it. Particularly when looking at air quality impact, this three-way comparison is of particular relevance as we generally expect a reduction in surface pollutants in future scenarios independently of SAI implementation (Fricko et al., 2017; Hussain, 2025; Nazarenko et al., 2022), therefore a comparison just between the present day and future SAI scenario will almost always indicate improved air quality. Therefore, comparing also the same future periods (which have the same surface emissions) with and without SAI helps to isolate the direct SAI contribution to air quality.

### 3.1 Changes in health-related air pollutants

Consistent with previous studies (Visioni et al., 2023), ARISE-SAI-1.5 exhibits an overall reduction in precipitation relative to the increase observed in SSP2-4.5 (Fig. 1f): these changes are due to both the avoidance of the temperature-related Clausius-Clapeyron increase expected under climate change, as well as to changes in the strength and position of the Intertropical Convergence Zone and the Hadley circulation (Kravitz et al., 2017; Lee et al., 2020; Richter et al., 2022; Cheng et al., 2022) in the different scenarios. While some regions do not exhibit statistically significant changes in surface $PM_{2.5}$ relative to SSP2-4.5 (2030–2039), other areas, such as Central America and central Sub-Saharan Africa, show statistically significant reductions. In these regions, $PM_{2.5}$ reductions coincide with increases in precipitation (Fig. 1d–f), suggesting that enhanced wet scavenging may play a role. However, the overall broader spatial pattern of $PM_{2.5}$ changes does not consistently track precipitation trends (Fig. 1g, h and j), and are not statistically significant, indicating that internal variability, circulation changes, vertical mixing, or aerosol-cloud interactions likely contribute to changes in $PM_{2.5}$. Thus, while precipitation influences $PM_{2.5}$ in some regions, it does not fully explain the spatial distribution or statistical significance of $PM_{2.5}$ changes, and many features of the $PM_{2.5}$ response should be interpreted with caution due to limited ensemble robustness.

Fig. 2 indicates that dust and secondary organic aerosols (SOA), rather than sulfate ($SO_4$), are the dominant contributors to total $PM_{2.5}$ concentrations across most regions in ARISE-SAI-1.5. Considering, however, that CESM2(WACCM6) is known to overestimate overall dust concentrations (e.g., Hancock et al., 2023), it is possible that this may contribute to dust appearing as the dominant $PM_{2.5}$ species in many regions, and this should therefore taken into account when interpreting Fig. 2. This potential upper bias does not affect the qualitative conclusion that sulfate is not the primary driver of $PM_{2.5}$ changes in our simulations, but it does mean that the relative prominence of dust should be interpreted with caution.

While some of the edges of the geographical ranges where each species dominates are stippled, indicating that fewer than 90% of ensemble members agree at the grid level, this ensemble uncertainty does not alter the overall conclusion that non-sulfate species dominate global $PM_{2.5}$. In SSP2-4.5 (not shown), the spatial distribution of the dominant $PM_{2.5}$ species is

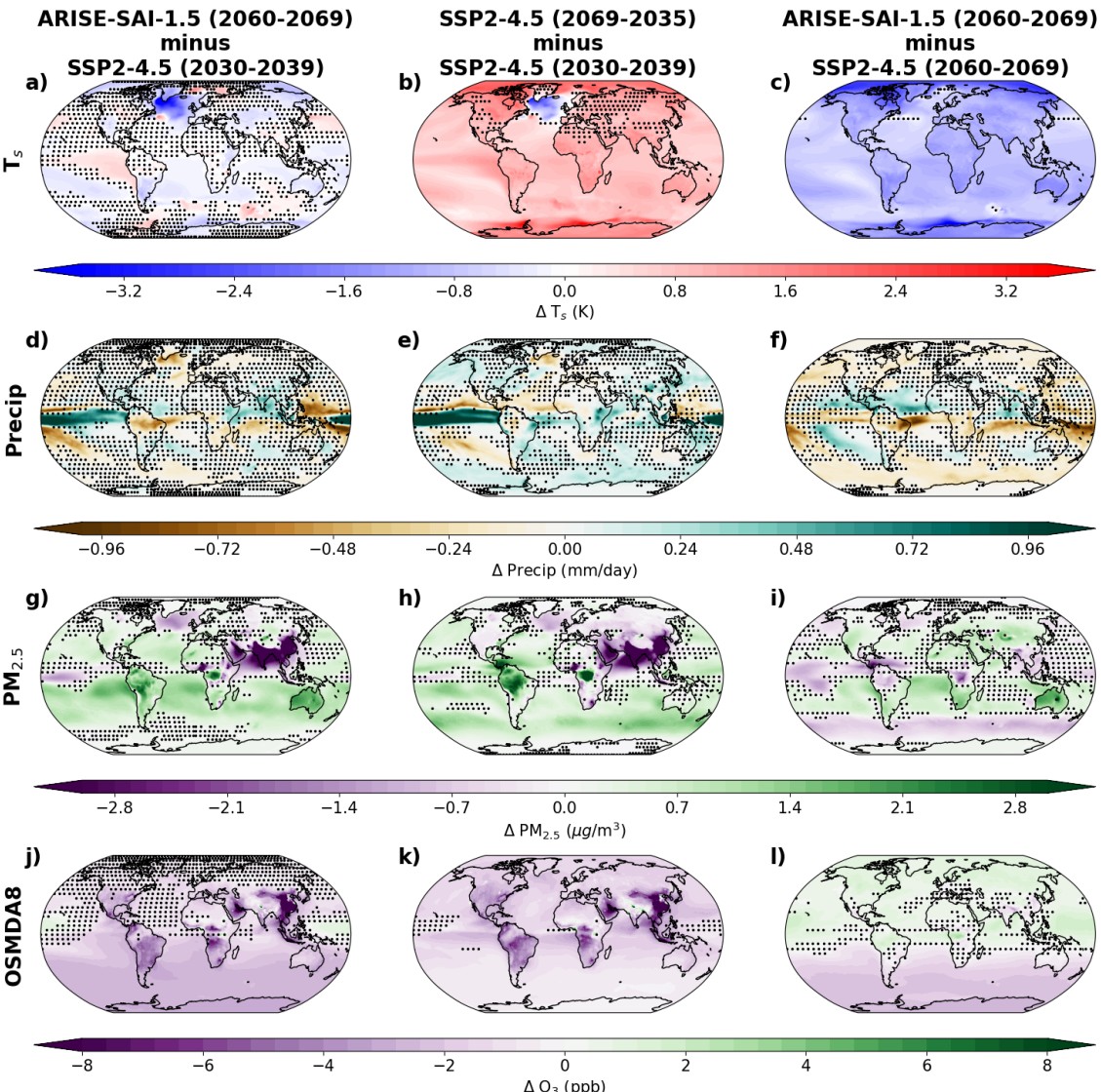

**Figure 1.** Spatial patterns of the changes in surface climate variables (surface temperature and precipitation), air quality (PM$_{2.5}$, and ozone exposure (OSMDA8) concentration) under the SAI scenario (ARISE-SAI-1.5) and the baseline scenario (SSP2-4.5) for the period 2060-2069 compared to 2030-2039. Each row represents changes in: (a-c) surface temperature (T$_s$, K), (d-f) precipitation (mm/day), (g-i) PM$_{2.5}$ concentration ($\mu g/m^3$), and (j-1) OSMDA8 (ppb). The stippling indicates areas where differences between ARISE-SAI-1.5 and SSP2-4.5 are not statistically significant (p>0.05) based on a t-test performed across all 10 ensemble members. Columns indicate the difference between the SAI case and the reference period with same global temperatures (left), the difference between a warmer future and the reference period (center), and the difference between the SAI case and a warmed future following the same underlying emission scenario (right).

broadly similar, with SO$_4$ not emerging as the dominant species across most regions, unless particularly pristine (Visioni et al., 2020), like at high latitudes, or already extremely polluted. While it is true that sulfate can still drive relative changes in PM$_{2.5}$ even when not dominant in absolute terms, our subsequent analysis of mortality (Section 3.2) shows that the changes in PM$_{2.5}$ concentrations and PM$_{2.5}$-related mortality are not driven by sulfate. Specifically, the spatial and temporal patterns of PM$_{2.5}$-related mortality changes align more closely with changes in non-sulfate species and are shaped by precipitation and circulation-driven effects such as wet scavenging and regional aerosol transport.

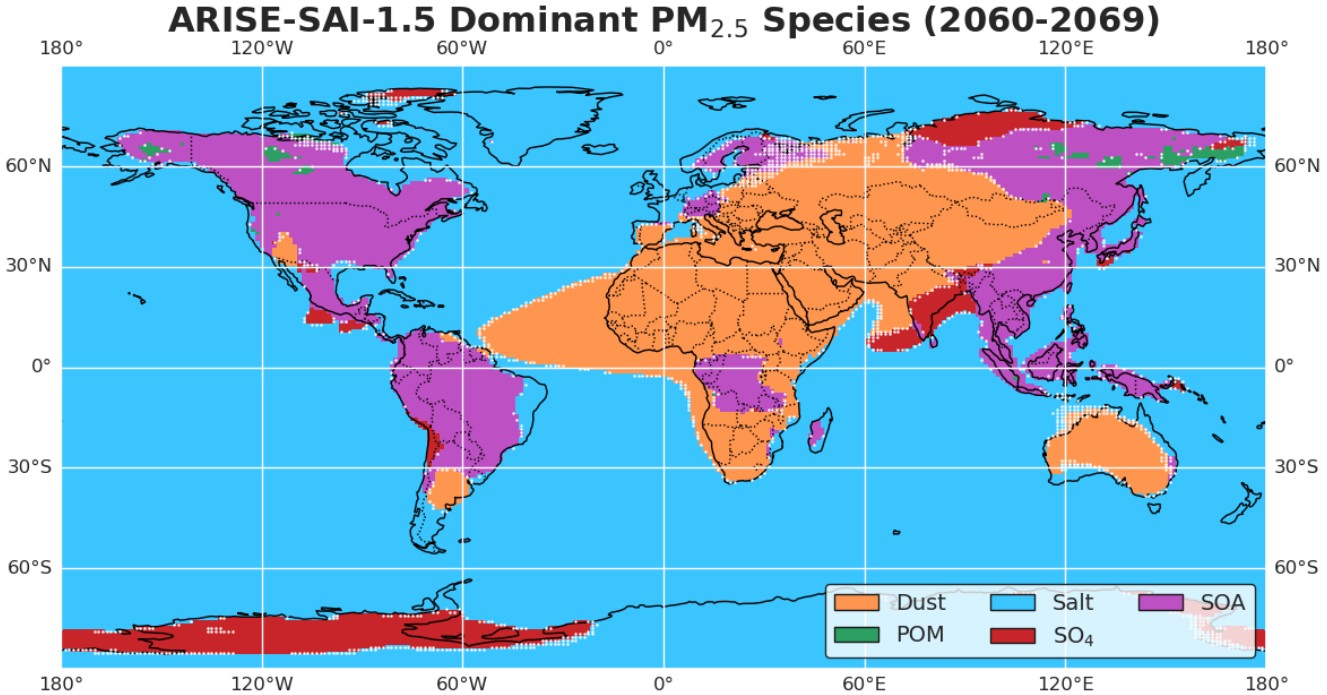

**Figure 2.** Map of the most prevalent PM$_{2.5}$ species (dust, primary organic matter (POM), salt, sulfate (SO$_4$), secondary organic aerosols (SOA) and black carbon (BC)) across grid cells, derived from ensemble model averages under the ARISE-SAI-1.5 scenario. Colors represent the dominant species at each location, determined by taking the fraction of the species to the total PM$_{2.5}$ concentration. Black carbon is not presented here because it does not dominate in any grid cell. White stippling is over areas where fewer than 90% of ensemble members agree on the dominant species at a grid point.

Fig. 1j-l shows % changes in surface ozone exposure. Interpreting these changes requires accounting for multiple mechanisms, including SAI-induced impacts on stratospheric ozone and its transport to the surface, and changes in ozone in-situ photochemical processing driven by changes in temperature and photolysis. SAI influences stratospheric ozone through multiple pathways, including alterations in heterogeneous chemical reactions on aerosol surfaces, modifications in photolysis rates due to changes in actinic flux from changes in the overhead ozone column and aerosol absorption and scattering, and dynamical changes in stratospheric circulation and temperature patterns that can impact ozone transport and distribution (Tilmes et al., 2009, 2022;

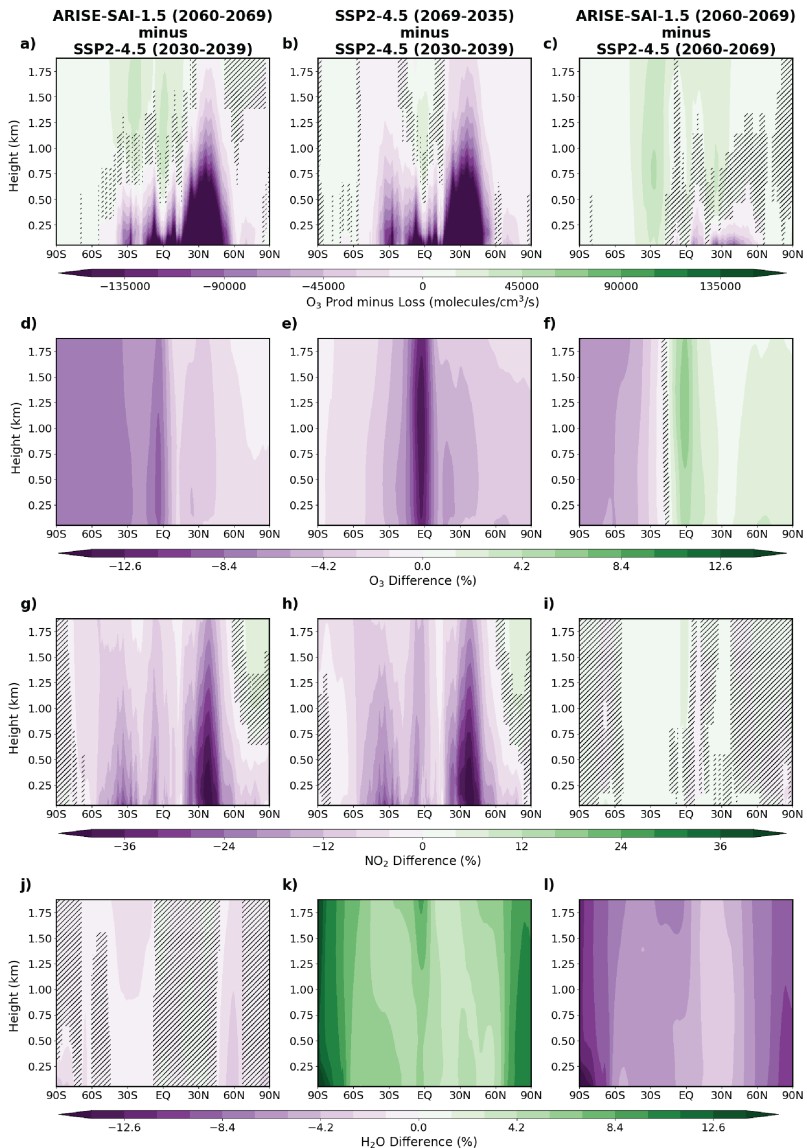

**Figure 3.** Zonal-mean % changes in ozone chemical production minus loss rates, ozone concentrations, and NO$_2$ concentrations under ARISE-SAI-1.5 and SSP2-4.5 scenarios. Panels (a–c) show the difference in ozone production minus loss (molecules/cm$^3$/s): (a) ARISE-SAI-1.5 (2060–2069) minus SSP2-4.5 (2030–2039), (b) SSP2-4.5 (2060–2069) minus SSP2-4.5 (2030–2039), and (c) ARISE-SAI-1.5 (2060–2069) minus SSP2-4.5 (2060–2069). Panels (d–f) show the corresponding % differences in ozone concentrations for the same scenario comparisons. Panels (g–i) show % differences in NO$_2$ concentrations, highlighting changes in a key ozone precursor and panels (j-l) show % differences in water vapor (H$_2$O) concentrations. The stippling indicates areas where differences between ARISE-SAI-1.5 and SSP2-4.5 are not statistically significant (p>0.05) based on a t-test performed across all 10 ensemble members.

Bednarz et al., 2023a). Injection strategy also plays a key role: in ARISE-SAI-1.5, $SO_2$ is injected primarily in the Southern Hemisphere (SH) during 2060–2069 to restore hemispheric temperature gradients affected by the asymmetric warming in the underlying SSP2-4.5 simulations, resulting in an asymmetric stratospheric aerosol burden and consequently an asymmetric ozone response (Richter et al., 2022; Bednarz et al., 2023b). However, as discussed before, our study does not include tropospheric chemistry changes caused by direct aerosol-driven changes in photolysis. As a result, our analysis does not capture potential tropospheric ozone responses caused by aerosol scattering (Visioni et al., 2017a). A study by Bardeen et al. (2021) using a previous version of WACCM (WACCM4), but modified to include online Tropospheric Ultraviolet and Visible (TUV) model calculations, showed that the exclusion of aerosol optical depth from the TUV calculations only resulted in a small difference in the overall ozone column changes due to minimal differences in the overall ozone loss rates, leading us to conclude that this shortcoming in our simulations is not likely to significantly impact our conclusions.

This stratospheric asymmetry propagates to the troposphere. Specifically, ozone concentrations exhibit a robust hemispheric asymmetry: decreases occur throughout much of the SH troposphere, while increases appear across the Northern Hemisphere (NH) (Fig. 1l and Fig. 3f). These hemispheric differences arise from distinct underlying mechanisms. In the SH, the reduction in surface ozone is primarily driven by aerosol-driven catalytic ozone loss in the Antarctic stratosphere alongside any changes in polar vortex strength and large-scale stratospheric transport (Bednarz et al., 2023b), and the resulting reduction in STE.

In contrast, the NH surface ozone increases are likely not driven by changes in STE. Although stratospheric ozone increases occur in the NH lower-to-mid stratosphere, this signal does not extend to the surface. Hence, the NH surface ozone changes likely reflect the SAI-induced changes in in-situ tropospheric chemical processing. In particular, water vapor concentrations decrease in the troposphere in ARISE-SAI-1.5 compared to SSP2-4.5 (Fig. 3l) as the result of large scale near-surface cooling (Fig. 1c). This reduces chemical ozone loss in the free-troposphere, as indicated by an increased net (i.e. production minus loss) photochemical ozone production (Fig. 3c). Due to rapid tropospheric mixing timescales, the resulting NH ozone increases extend to the surface, even despite negative (particularly between 0 to 50°N) NH surface net production changes under SAI. The latter indicate suppressed in-situ photochemical ozone formation that occurs in a $NO_x$-rich region (Fig. 3g and h) under decreased OH and the resulting suppressed $RO_2$-$NO_2$ cycling (despite a concurrent increase in NH surface $NO_x$ which should otherwise enhance ozone production, Fig. 3i), consistent with previous work demonstrating that reductions in temperature and humidity can suppress photochemical ozone formation in $NO_x$-rich environments (Archibald et al., 2020; Rasmussen et al., 2013; Doherty et al., 2013; Zanis et al., 2022).

To further test this interpretation, we repeated our analyses in simulations with simulated SAI injections but no changes in tropospheric anthropogenic emissions (i.e. in a preindustrial climate) and observed qualitatively similar ozone responses (not shown), reinforcing our finding that these changes arise from stratospheric chemistry, transport, and in-situ oxidant perturbations, consistent with previous findings on SAI-driven ozone redistribution (e.g., Xia et al., 2017; Niemeier and Schmidt, 2017; Tilmes et al., 2009).

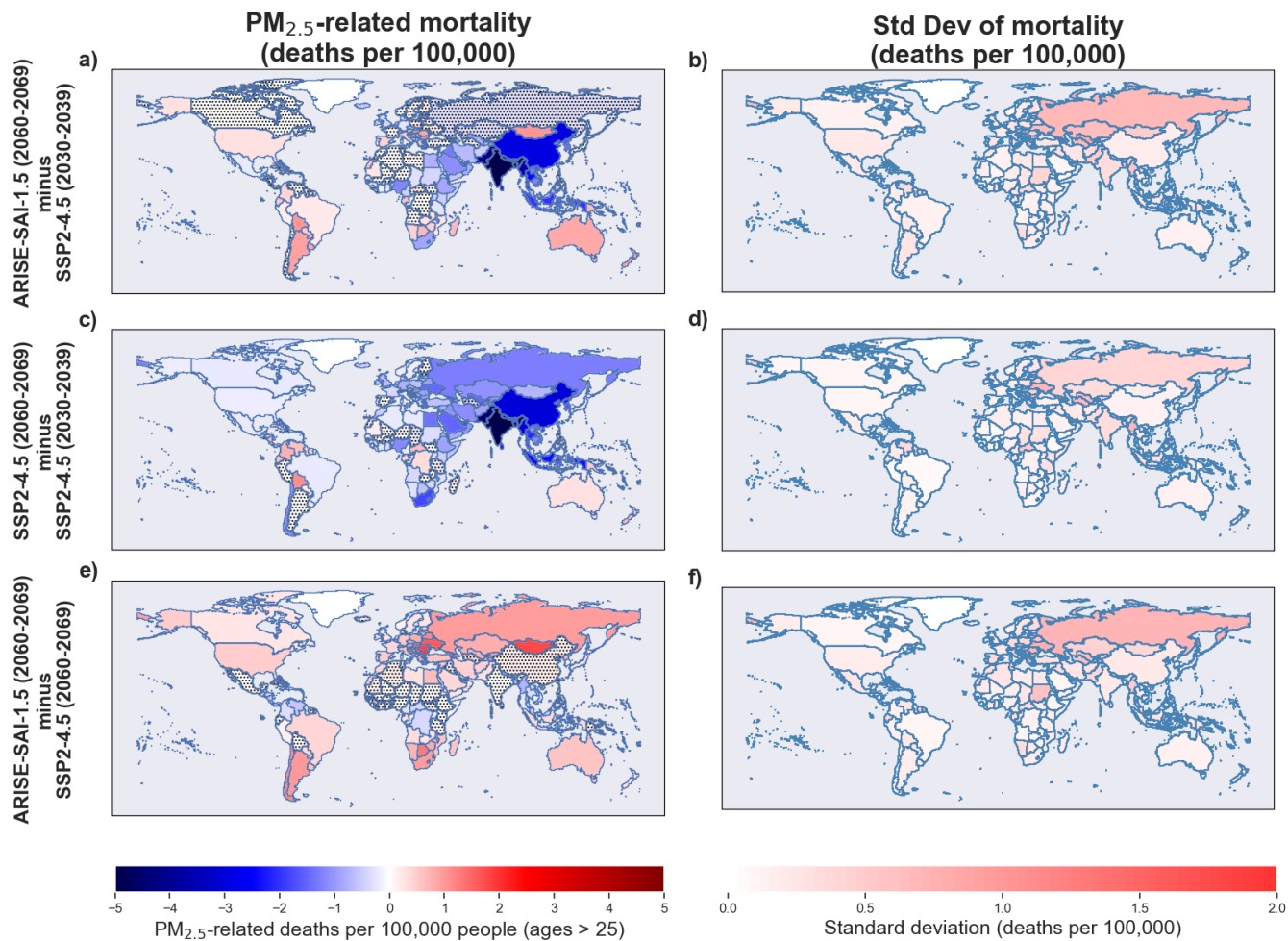

**Figure 4.** Country-level PM$_{2.5}$-related mortality changes (per 100,000 people) based on ten-member CESM2(WACCM6) ensembles for ages > 25. Each row shows (left) the ensemble-mean mortality change and (right) the inter-ensemble standard deviation. Row (a–b) shows ARISE-SAI-1.5 for 2060–2069 relative to SSP2-4.5 for 2030–2039; (c–d) shows SSP2-4.5 for 2060–2069 relative to its own 2030–2039 baseline; and (e–f) shows the direct effect of SAI by comparing ARISE-SAI-1.5 to SSP2-4.5 for the same 2060–2069 period. Mortality is normalized by national population totals and reported as deaths per 100,000 people (ages > 25).Stippling indicates countries where estimates of the PM$_{2.5}$-related mortality are not statistically significant across ensemble members at the 95% confidence level.

## 3.2 Calculation of the air pollution related mortality from PM$_{2.5}$ and ozone changes

This section presents the estimated mortality impacts of SAI under the ARISE-SAI-1.5 protocol, relative to SSP2-4.5. We first examine changes in PM$_{2.5}$-related mortality resulting from SAI, followed by an assessment of ozone-related mortality. Together with showing ensemble-averaged results, we also highlight in the following maps the large inter-ensemble and inter-ensemble spread when calculating mortality based on yearly model output. Local air quality is strongly dependent on meteorological

conditions (Liu et al., 2022; Jacob and Winner, 2009; Xu et al., 2020) such as precipitation rates, heatwaves and atmospheric
inversions. Global warming itself has been postulated to strengthen many of these conditions as well (Jacob and Winner, 2009).
Therefore, it is important to interpret our estimates within this broader context.

Fig. 4 and 7a show the annual global deaths resulting from changes in $PM_{2.5}$ concentration and the average $PM_{2.5}$-related
deaths (per 100,000 people) by country, respectively. Globally, we estimate that SAI leads to a reduction of ~151,000 premature
deaths from $PM_{2.5}$ under ARISE-SAI-1.5 (2060-2069), relative to SSP2-4.5 (2030-2039), with ensemble member estimates
ranging from -140,000 to
-164,000. In comparison, SSP2-4.5 (2060-2069) results in a reduction of ~165,000 premature deaths relative to 2030-2039
levels, with a range of -148,000 to -177,000. This yields a net increase of ~14,000 premature deaths in ARISE-SAI-1.5
compared to SSP2-4.5 during 2060-2069, with an ensemble range of -7,000 to +21,000. These estimates, along with the
standard deviation shown in Fig. 4, illustrate the substantial variability in projected $PM_{2.5}$-related deaths.

The changes in $PM_{2.5}$-related mortality for each country in Fig. 4e are roughly consistent with the geographical changes in
$PM_{2.5}$ shown in Fig. 1i. In particular, the $PM_{2.5}$-related changes in certain countries (e.g., Russia and several in Sub-Saharan
Africa) primarily reflect regions where internal variability, rather than an SAI-driven signal, dominates–consistent with the
broader spatial pattern of $PM_{2.5}$ that are not statistically significant (Fig. 1i). In Fig. 5a, we compute the ensemble-averaged
global deaths resulting from SSP2-4.5 with added changes in individual $PM_{2.5}$ components between ARISE-SAI-1.5 and SSP2-
4.5 to isolate the influence of each component on global mortality. Among the components, incorporating changes in the dust
$PM_{2.5}$ produce the largest deviation from the unmodified SSP2-4.5 baseline. The scenario with dust-only modifications results
in fewer global deaths than the SSP2-4.5 baseline, which is likely due to the nonlinearity in the ozone-attributable risk function.
However, when changes in all $PM_{2.5}$ components are combined, the resulting mortality aligns with the increased PM2.5-related
mortality observed in ARISE-SAI-1.5. For other components such as salt, BC, POM, SOA and $SO_4$, the resulting mortality
estimates largely overlap the unmodified SSP2-4.5 baseline. In particular, the changes in global deaths attributable to $SO_4$
are small relative to other components, implying that sulfate-driven $PM_{2.5}$ mortality changes are modest compared to the
total. Therefore, we conclude that SAI's contribution to $PM_{2.5}$–related mortality is small compared to the overall changes
projected due to future air quality policies (on the order of ~1%, versus ~10% from policy-driven improvements), with
internal variability among ensemble members and changes from other $PM_{2.5}$–related species playing a dominant role in driving
uncertainty in our mortality estimates.

For ozone-related mortality, Fig. 6 and 7b show the annual global total deaths resulting from changes in ozone concentration
and the average ozone-related deaths by country, respectively. We estimate that SAI leads to a reduction of ~102,000 premature
deaths from ozone exposure under ARISE-SAI-1.5 (2060–2069), relative to SSP2-4.5 (2030–2039), with an ensemble range
of -91,000 to -108,000. By comparison, SSP2-4.5 (2060-2069) results in an estimated reduction of ~89,000 premature deaths
from ozone exposure relative to SSP2-4.5 for 2030–2039, with a range of -77,000 to -97,000. The net difference between
ARISE-SAI-1.5 and SSP2-4.5 during 2060-2069 is ~-14,000, with a range of -7,000 to -25,000.

The geographic distribution of ozone-related mortality changes (Fig. 6e) shows that the largest reductions occur primarily
in the Southern Hemisphere, most notably over Southern Sub-Saharan Africa, Southeast Asia and South America. This spatial

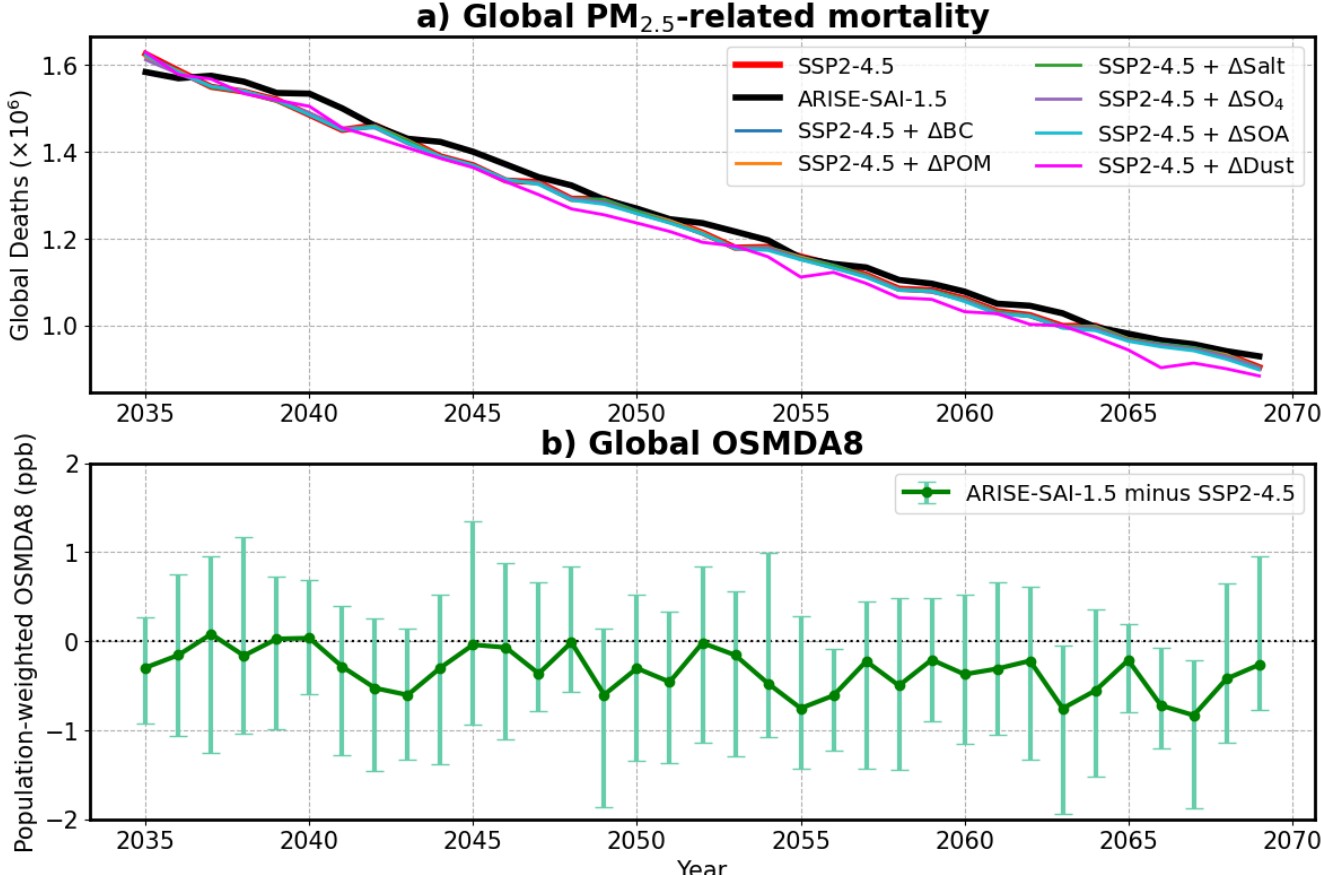

**Figure 5.** (a) Ensemble-averaged global PM$_{2.5}$-related mortality over time under the SSP2-4.5 and ARISE-SAI-1.5 scenarios, along with sensitivity simulations where changes in individual PM$_{2.5}$ components ($\Delta$BC, $\Delta$POM, $\Delta$Salt, $\Delta$SO$_4$, $\Delta$SOA, $\Delta$Dust) between ARISE-SAI-1.5 and SSP2-4.5 are added to the SSP2-4.5 baseline. (b) Time series of population-weighted global OSMDA8 (daily maximum 8-hour ozone) differences between ARISE-SAI-1.5 and SSP2-4.5, with error bars indicating ensemble spread.

pattern aligns with the hemispheric asymmetry in the tropospheric ozone response observed in Fig. 3, where greater reductions in ozone concentrations occur in the SH and parts of Asia. In Fig. 7b, the evolution of global ozone-related deaths over time is consistent with the time series of global OSMDA8 (Fig. 5b). Overall, no clear long-term trend is evident in PM$_{2.5}$ and ozone-related mortality, as any underlying signal may be masked by the large ensemble variability in projected deaths (Fig. 7). Geographically, both ozone-and PM$_{2.5}$-related mortality changes exhibit substantial spatial variability, driven by regional differences in how ozone and PM$_{2.5}$ concentrations respond to shifts in atmospheric chemistry, circulation, and precipitation patterns under SAI.

Figure 7 shows how global changes in mortality due to ozone and PM$_{2.5}$ evolve over time in our simulations. When aggregated globally, it is evident that the largest change in air-pollution related mortality is due to decreases in precursors and

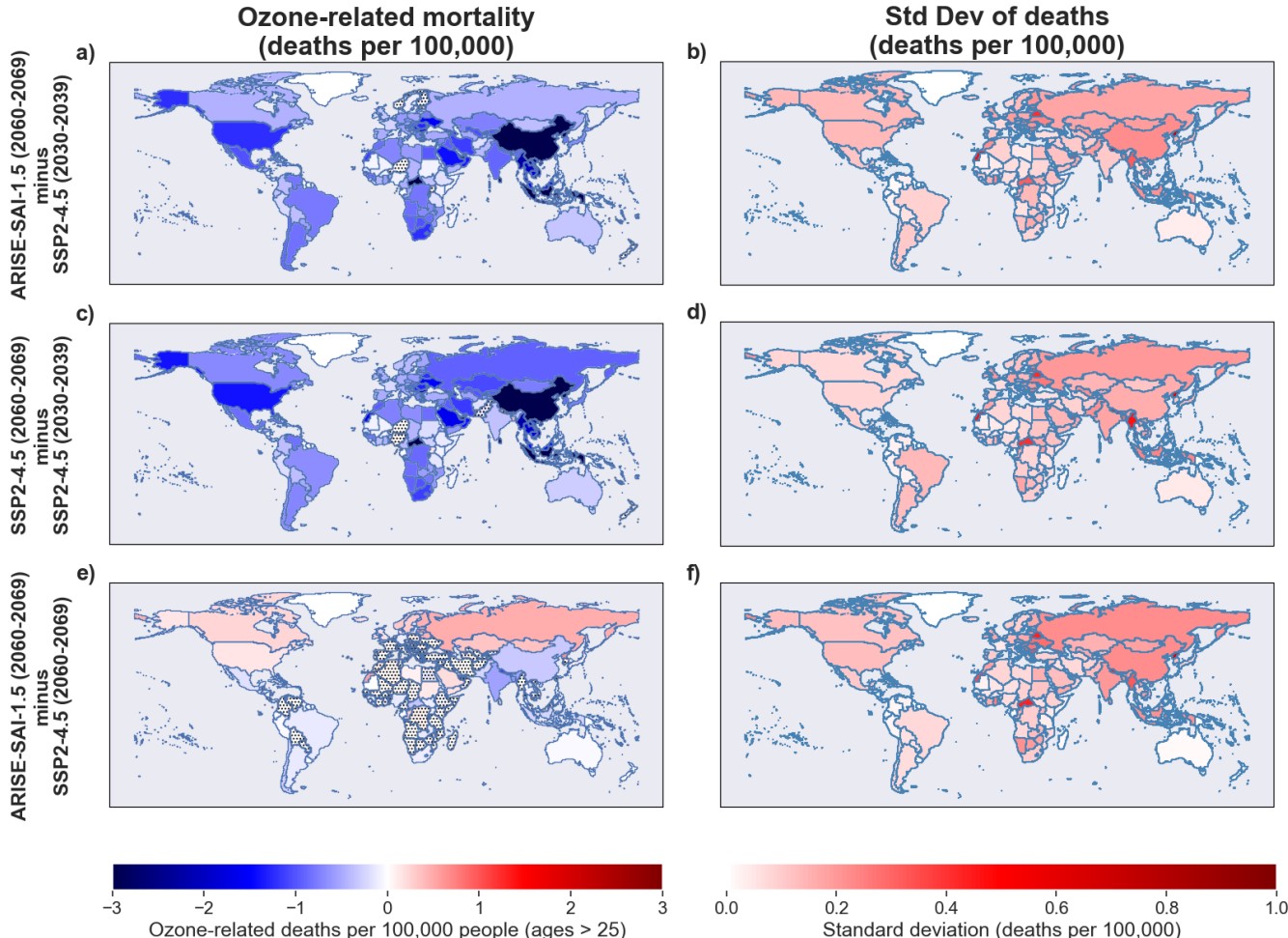

**Figure 6.** Country-level ozone-related mortality changes (per 100,000 people) for ARISE-SAI-1.5 and SSP2-4.5 simulations. Each row shows the ensemble-mean mortality change (left column) and the corresponding inter-ensemble standard deviation (right column) across ten CESM2(WACCM6) ensemble members. Row (a–b): ARISE-SAI-1.5 for 2060–2069 relative to SSP2-4.5 for 2030–2039; (c–d): SSP2-4.5 for 2060–2069 relative to its own 2030–2039 baseline; (e–f): ARISE-SAI-1.5 relative to SSP2-4.5 for the same 2060–2069 period, isolating the direct SAI effect. Mortality values are normalized by total population in each country and expressed in deaths per 100,000 people. Stippling indicates countries where estimates of the ozone-related mortality are not statistically significant across ensemble members at the 95% confidence level.

pollutants under the SSP2-4.5 scenario. Differences between the futures with and without SAI, and those between different amount of SAI cooling, are much smaller on a per-year basis, and in most cases within the range of variability for the ensemble estimates. This demonstrates that the direct impact of deposited sulfate is limited, and climatic factors minimally impact $PM_{2.5}$ changes under SAI. In contrast, global ozone-related mortality is slightly lower in ARISE-SAI-1.0 than in ARISE-SAI-

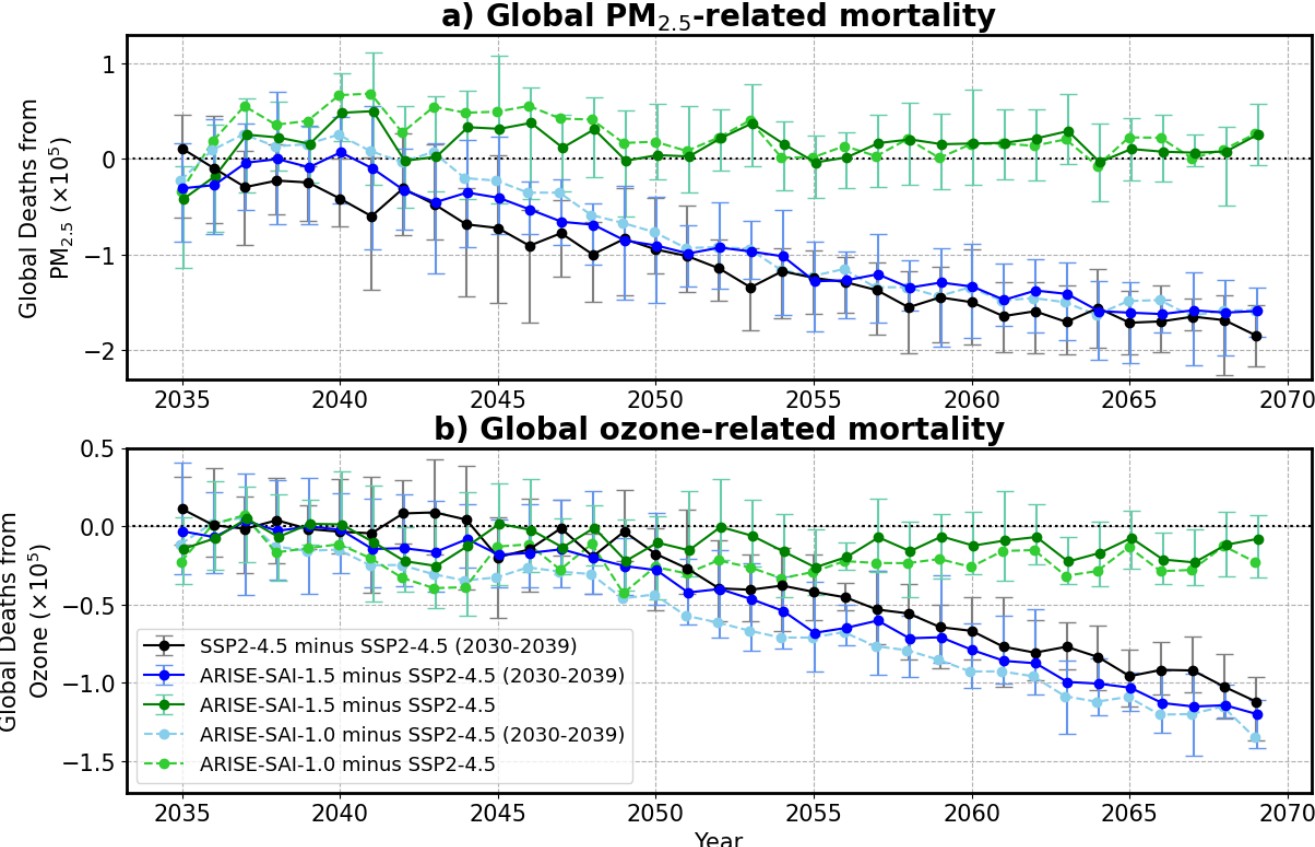

**Figure 7.** Global deaths from a) PM$_{2.5}$ and b) ozone evaluated as 1) ARISE-SAI-1.O minus SSP2-4.5, 2) ARISE-SAI-1.5 minus SSP2-4.5 (2030-2039), 3) ARISE-SAI-1.5 minus SSP2-4.5, 4) SSP2-4.5 minus SSP2-4.5 (2030-3029) and 5) ARISE-SAI-1.5 minus SSP2-4.5 (2030-2039). Error bars represent the full range of outcomes across the model ensemble, showing the minimum and maximum values.

1.5, likely due to larger SAI-induced SH extra-tropical lower stratospheric ozone loss and the resulting reduction in ozone stratosphere-troposphere transport.

Figure 8 provides an alternative way of examining this relationship by plotting ensemble means against injection rates in the SAI scenarios. For both simulations, PM$_{2.5}$-related mortality shows no clear linear scaling with increasing injection rates, as substantial ensemble variability and factors other than SAI affecting the evolution of mortality rates with time dominate the relationship. Ozone-attributable mortality remains consistently negative across the entire injection range, indicating a reduction in ozone-related deaths under both ARISE-SAI-1.5 and ARISE-SAI-1.0.

For PM$_{2.5}$-related mortality in particular, our component attribution analysis suggests that the primary driver of changes is not sulfate itself, but rather arises from changes in dust and secondary SOA concentrations (Figs. 2 and 5a). Regional reductions in PM$_{2.5}$, particularly over Central America and central Sub-Saharan Africa, align with areas of increased precipitation,

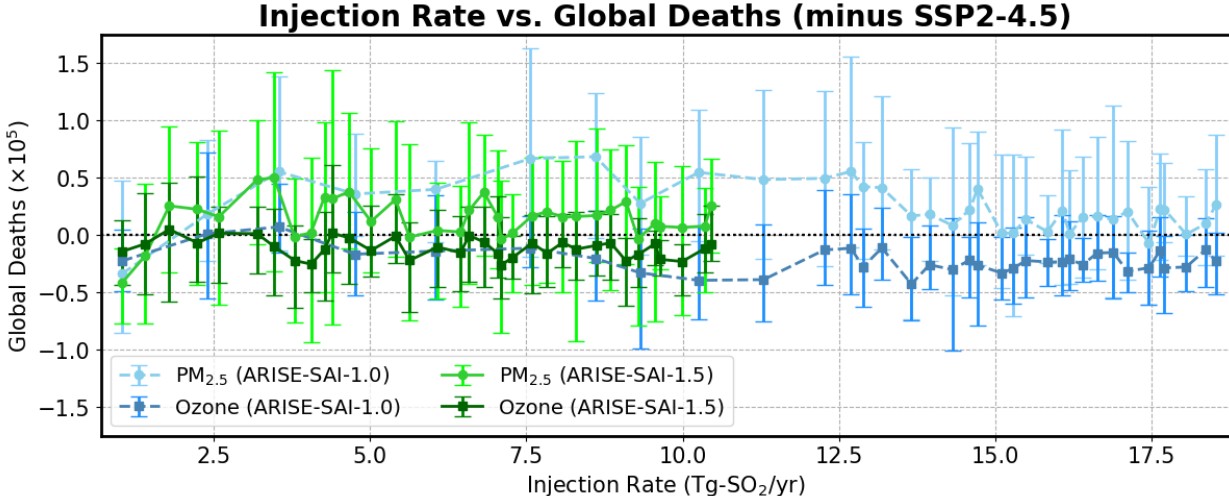

**Figure 8.** Global mortality differences for ARISE-SAI-1.5 minus SSP2-4.5 (shades of green) and ARISE-SAI-1.0 minus SSP2-4.5 (shades of blue) as a function of annual $SO_2$ injection rate (Tg-$SO_2$/yr), for $PM_{2.5}$-related deaths (green) and ozone-related deaths (green). Points represent ensemble means across years (2035–2069), with error bars indicating ensemble variability ($\pm 1$ standard deviation).

highlighting the role of wet deposition and circulation-driven suppression of natural aerosol sources (Fig. 1). However, the widespread lack of statistically significant precipitation or $PM_{2.5}$ changes across ensemble members suggests that internal variability and regional circulation shifts, rather than sulfate burden alone, govern the spatial and temporal patterns of $PM_{2.5}$-
related health outcomes under SAI.

For ozone, the mortality reductions appear more discernible. $SO_2$ is primarily injected in the SH, leading to decreased SH extra-tropical lower stratospheric ozone concentrations and the resulting reduction in SH surface ozone from reduced STE overwhelming any in-situ changes in tropospheric ozone chemistry there. In the NH, on the other hand, surface ozone increases due to the suppressed photochemical destruction under drier and colder troposphere (Fig. 3). These changes reflect the role
of not only hemispheric asymmetries in sulfate burden alone but also those in STE and chemical processing arising from circulation changes and altered chemical regimes in shaping global ozone responses and associated health outcomes under SAI.

Taken together, these findings emphasize that air pollution-related health impacts under SAI are not governed mainly by the magnitude of $SO_2$ injected, but rather by the complex suite of dynamical, chemical, and aerosol responses in the Earth system–
many of which are nonlinear and strongly influenced by internal variability. While our two large ensemble SAI simulations show no evidence for linear scaling with respect to injection rate, we acknowledge that longer simulations and additional scenarios would be needed to more fully characterize how air quality related mortality is dependent on the SAI scenario under consideration.

### 3.3 Global Burden of Disease super-region specific projections

Globally, ARISE-SAI-1.5 reduces total pollution-attributable mortality relative to a future without intervention (SSP2-4.5) by 0.4%, driven by a 0.9% increase in $PM_{2.5}$ and 1.3% reduction in ozone-related deaths (Fig. 9a–b). However, the direction and magnitude of health outcomes vary substantially across GBD super-regions. For instance, large % increases in $PM_{2.5}$-related mortality occur in regions such as Central, Western and Eastern Europe. In contrast, regions like the Caribbean and Central Latin America exhibit reductions in $PM_{2.5}$-attributable mortality, highlighting the heterogeneous and sometimes
adverse regional impacts of SAI.

For ozone-related mortality, the ensemble spread is also large–both in magnitude and spatial extent–especially in regions such as the Western and Eastern Sub-Saharan Africa and the Caribbean. Furthermore, while national base mortality (cardiovascular, respiratory, and NCD+LRI baseline mortality rate) declines from 2030–2039 to 2060–2069 across all regions, the magnitude of these changes is relatively small compared to the much larger shifts seen in air quality-related mortality.

In many regions, the large ensemble spread reflects uncertainties not only in the magnitude but also in the sign of the projected impact on air quality related mortality. This spread arises from internal climate variability, which influences key drivers of air quality–such as atmospheric circulation, precipitation patterns, and chemical processing–and leads to diverging pollutant concentrations across ensemble members, even under identical forcing scenarios. These findings also highlight the spatial heterogeneity in health responses to SAI. While global or hemispheric trends may point to a net decline in ozone-
related mortality and an increase in $PM_{2.5}$-related mortality, such aggregates can mask substantial regional disparaties. As a result, careful evaluation of region-specific trade-offs is critical when assessing the overall public health implications of SAI deployment.

### 4  Conclusions

This study evaluates the impacts of SAI on air quality–related mortality using a fully coupled climate model ensemble
under the ARISE-SAI protocol. Unlike previous studies using CTMs (e.g., Eastham et al. 2018; Moch et al. 2023), which imposed stratospheric aerosols without capturing feedbacks on dynamics and transport, our use of CESM2(WACCM6) enables interactive coupling between aerosols, chemistry, and climate.

We explore two potential sources of mortality: $PM_{2.5}$ and surface ozone exposure. $PM_{2.5}$ is affected both by direct deposition of sulfate from the stratosphere and by climatic conditions affecting other sources of particulates. Tropospheric $O_3$ changes
from SAI can be driven by the combination of changes in stratospheric ozone and its transport to the troposphere, and by in-situ changes in tropospheric ozone chemistry driven by SAI-induced changes in surface temperatures and photolysis. The latter are not fully considered in our study, as tropospheric ozone changes caused by direct aerosol impacts on actinic fluxes and photolysis rates are not included (although photolysis rates would still be affected indirectly by aerosol-driven changes to stratospheric ozone column above, and by cloud changes).

We find that the direct contribution of sulfate aerosols to $PM_{2.5}$-related mortality is minimal, primarily because much of the injected sulfate is transported poleward and deposited at mid-latitudes, leading to a relatively diffuse and spatially uniform

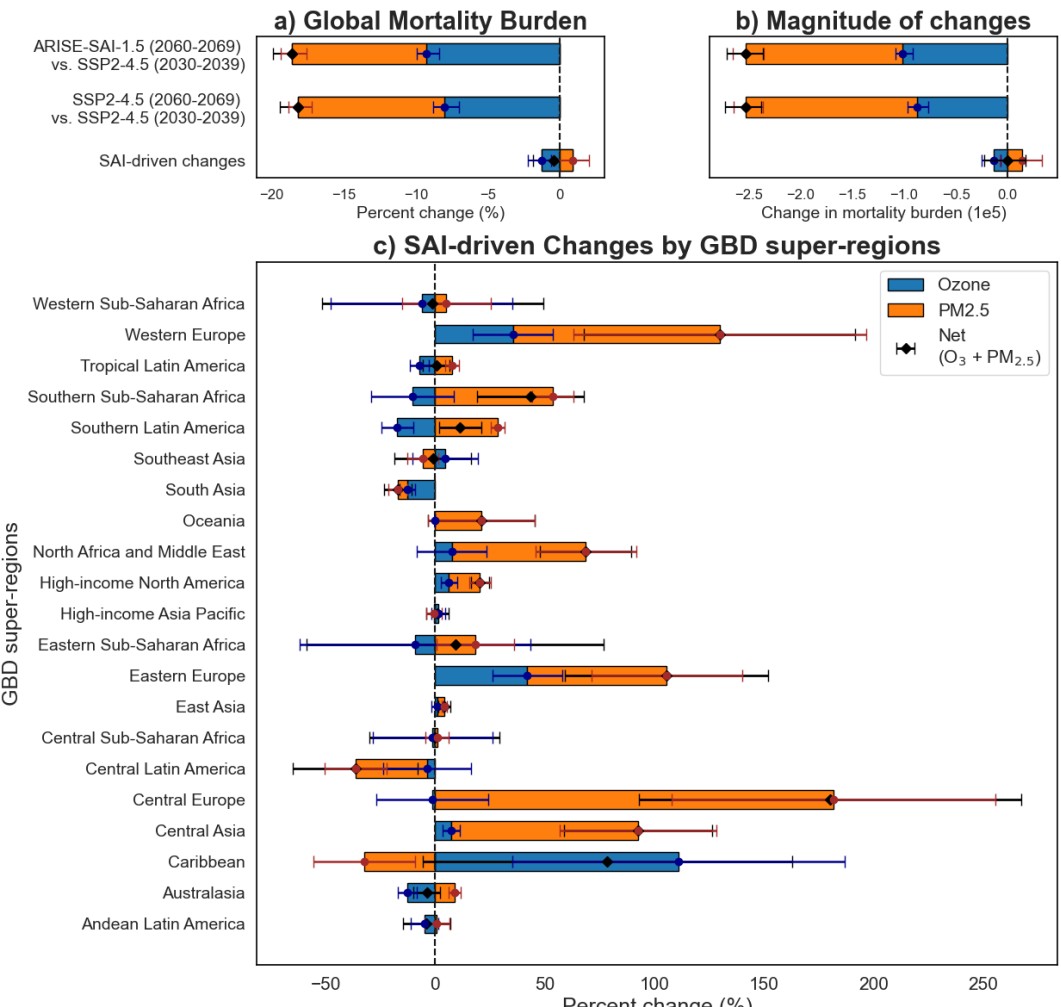

**Figure 9.** (a) Global % change in mortality burden comparing ARISE-SAI-1.5 (2060–2069) and SSP2-4.5 (2030–2039), as well as SSP2-4.5 (2060–2069) and SSP2-4.5 (2030–2039). (b) Absolute global changes in mortality burden (in number of deaths). (c) % change in mortality burden by Global Burden of Disease (GBD) super-region between ARISE-SAI-1.5 (2060-2069) and SSP2-4.5 (2060-2069). % changes are calculated relative to baseline mortality rates. Positive values indicate an increase in mortality relative to the baseline, while negative values indicate reductions. Bars represent stacked contributions from ozone-related deaths (blue) and $PM_{2.5}$-related deaths (orange), with horizontal error bars indicating the ensemble spread (standard deviation) for each component and for the net total (black diamonds with error bars).

distribution. Furthermore, a portion of sulfate particles exceed the $PM_{2.5}$ size threshold and therefore does not contribute to fine particulate mass. Subsequently, the total mass of sulfate aerosols reaching the surface is insufficient to meaningfully alter concentration thresholds associated with mortality outcomes. Instead, regional changes in $PM_{2.5}$ concentrations and the 375 corresponding health impacts are mainly driven by shifts in precipitation patterns and/or circulation, which affect the wet

removal of non-sulfate species such as dust and secondary organic aerosols, consistent with Eastham et al. (2018). Likewise, we find that ozone-related mortality is projected to maintain its decrease globally due to changes in pollutant sources even under SAI; but, when comparing the two future scenarios, the SAI impact result in a change in the spatial pattern reflecting a hemispheric asymmetry in the tropospheric ozone response, leading to a slight increase in surface ozone in the NH and a

decrease in the SH. However, some uncertainties related to the specific evolution of surface ozone remain, particularly due to the absence of the direct aerosol effect on the photolysis rates, which could lead to an underestimation of chemical feedbacks in the troposphere.

All mortality estimates in our future scenarios are calculated using the fixed 2020 population distribution. This approach isolates the effects of air quality changes by removing confounding influences from projected population growth or redistribution.

However, mortality rates could be significantly affected by demographic and population changes, such as aging, urbanization, or overall population growth, which are not considered in this study. As a result, our estimates may not fully reflect future health impacts under evolving demographic conditions.

Furthermore, our analysis is based on a single climate model and two closely related SAI scenarios, and thus the results may be both scenario and model-dependent. However, comparisons between ARISE-SAI-1.0 and ARISE-SAI.1.5 indicate

that global $PM_{2.5}$-related mortality does not increase significantly under higher $SO_2$ injection amounts, whereas ozone-related mortality decreases sightly with higher injection rates due to lower temperatures. This suggests that variability in PM2.5-related mortality may be more strongly influenced by changes in dust or biomass-burning-derived PM2.5 driven by circulation responses to SAI, rather than directly by the total amount of $SO_2$ injected. However, because $SO_2$ is primarily injected in the SH for these scenarios, it may also be relevant to examine whether similar observations emerge under a broader set of scenarios.

Future assessments of SAI impacts on air quality and related mortality could be improved by multi-model intercomparisons to better constrain the contributions of non-sulfate aerosol species, such as dust, BC, and SOA, as well as to capture the range of model uncertainty in aerosol-chemistry climate interactions. Additionally, improved representation and observational verification of large-scale circulation responses, particularly changes in the Brewer-Dobson Circulation and STE, are essential for understanding the transport and distribution of injected aerosols, as well as their downstream effects on regional air quality.

Furthermore, the incorporation of more detailed aerosol microphysics, including size-resolved coagulation, nucleation, and heterogeneous chemistry, would allow for a more accurate simulation of aerosol growth, lifetime, and radiative properties. Together, these efforts would enable more comprehensive and policy-relevant evaluations of SAI's atmospheric and health impacts.

While this study focused on the air quality-related health impacts of SAI, it is important to acknowledge that other health-

relevant outcomes, such as changes in surface UV radiation and regional temperatures, were not evaluated here but may also carry significant implications. Preliminary analysis of surface UV radiation differences between ARISE-SAI-1.5 and SSP2-4.5, calculated with the Tropospheric Ultraviolet and Visible (TUV) model developed at NCAR (Madronich and Flocke, 1999; Visioni et al., 2017b), show that changes in surface UV are small (between -5.3 to -6.1% globally). This finding is broadly consistent with previous studies that examined UV responses to SAI, including recent work highlighting that while

atratospheric aerosol perturbations can modify photolysis rates, the net surface UV changes tend to be modest (Bardeen

et al., 2021). Although small, such changes could still influence surface ozone through altered photochemistry and may affect secondary particulate matter, such as POM and SOA, by modifying photolysis-driven oxidation pathways. These potential impacts remain an important avenue for future investigation.

In addition, other processes known to affect air quality under climate change, such as changes in planetary boundary layer height (Deng et al., 2023; Li et al., 2017, 2019) and lighting activity (Murray, 2016; Grewe, 2009) could play a significant role to the simulated air quality response to SAI. However, the scope of this paper is to assess the net outcome of these combined processes for surface-level $PM_{2.5}$ and ozone concentrations, and their associated health effects, across large ensembles. As with air quality, they are part of a broader suite of SAI-induced environmental changes that warrant further exploration.

Internal climate variability plays a critical role in modulating aerosol transport, chemical processes, regional temperature responses, and stratospheric ozone dynamics. By resolving dynamic feedbacks between aerosols, transport, and atmospheric chemistry, our modeling approach overcomes key limitations of earlier CTM-based studies, enabling more realistic estimates of SAI-induced air quality and health outcomes. This highlights the importance of using fully coupled Earth system models when evaluating the policy-relevant consequences of geoengineering strategies and reinforces the need to account for natural variability when assessing human health impacts. Our results, which emphasize the importance of ensemble approaches for air pollution mortality estimates, highlight a general need for robust ensemble-based evaluations across all dimensions of SAI's potential risks and trade-offs.

When viewed in the context of climate change impacts on air quality, our findings suggest that the additional effects of SAI are small relative to both internal variability and policy-driven improvements. Prior studies have identified a "climate penalty" on air quality, in which rising temperatures and shifts in meteorology under climate change can increase surface ozone and fine particulate concentrations, resulting in increases in air pollution-related mortality (Fiore et al., 2015; Doherty et al., 2013; Fu and Tian, 2019; Silva et al., 2017). SSP2-4.5 represents a moderate mitigation and policy pathway, in which partial greenhouse gas reductions are achieved, leading to some reductions in $CO_2$, $CH_4$, and co-emitted air pollutants, and consequently modest improvements in air quality relative to higher-emission futures (Hussain, 2025; Nazarenko et al., 2022; Shim et al., 2021). In our simulations, SSP2-4.5 leads to an 18% (ensemble range: -19 to -17%) reduction in air pollution-related mortality relative to present day (2030-2039), driven primarily by emissions policies. Under ARISE-SAI-1.5, mortality is reduced by a similar amount (19%; -20 to -18%), with the net impact of SAI largely falling within the range of internal variability. This finding highlights that while SAI can shift the spatial distribution of ozone and particulate matter, particularly through hemispheric asymmetries in stratospheric aerosol loading and associated dynamical responses, the dominant driver of future health outcomes remains the strength of air quality policies (Vandyck et al., 2018). Our results therefore align with the broader literature emphasizing that while internal variability can obscure the precise effects of climate change (Pienkosz et al., 2019; Garcia-Menendez et al., 2017) and even climate interventions on air quality, sustained emissions reductions are important in determining future air quality and health outcomes.

. Code used in computing the $PM_{2.5}$ and ozone-related mortality can be found at https://doi.org/10.5281/zenodo.15696232 (Wang, 2025). All the data presented in this paper are available at https://doi.org/10.5281/zenodo.6473954 (Richter and Visioni, 2022b) from the CESM2(WACCM6)

SSP2–4.5 simulations and at https://doi.org/10.5281/zenodo.6473775 (Richter and Visioni, 2022a) from the ARISE-SAI simulations.

. CW performed the analysis of the model simulations and wrote the manuscript. DV helped outline and write the manuscript and provided insight on the analyses. GC provided guidance on how to conduct the mortality calculations and helped write the manuscript. EB provided insight on the analysis and contributed to the writing of the manuscript.

. At least one of the (co-)authors is a member of the editorial board of Atmospheric Chemistry and Physics.

. CW and DV acknowledge financial support by the Quadrature Climate Foundation. We thank Douglas G. MacMartin for his constructive discussions and suggestions during the development of this work. GC is supported by an appointment to the NASA Postdoctoral Program at the Goddard Institute for Space Studies, administered by Oak Ridge Associated Universities under contract with NASA. We would also like to thank Yutang Xiong and Collin Meisel at the Frederick S. Pardee Institute for International Futures for their help with accessing the IFS platform and quick response to inquiries. We would like to acknowledge high-performance computing support from the Derecho system

(doi:10.5065/qx9a-pg09) provided by the NSF National Center for Atmospheric Research (NCAR), sponsored by the National Science Foundation.

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
