# Peer review of "Air quality impacts of stratospheric aerosol injections are likely small and mainly driven by changes in climate, not aerosol settling"

_EGUsphere, 2025_

## Referee Comment (RC1)

[referee-annotated manuscript omitted]

---

## Author Comment (AC1)

Reviewer comments are in **bold** and the authors' responses are in blue.

We thank the reviewer for their thorough and constructive evaluation of our manuscript. The comments have been invaluable in helping us clarify the scope of our study, better communicate the limitations of our modeling framework, and improve the presentation of the results. In revising the manuscript, we have explicitly noted caveats regarding missing processes in the model (e.g., fixed tropospheric photolysis, absence of ammonium/nitrate aerosols), clarified which conclusions apply specifically to the ARISE scenarios, and removed language that overstated the comprehensiveness of the model. We have also improved the abstract and figures, added references and explanations where needed, and corrected wording and formatting issues. We have also implemented many of the text, wording, and grammar edits that have been suggested by the reviewer. We appreciate the reviewer's concern regarding the length of the manuscript. While the overall message may appear conceptually straightforward, arriving at this conclusion requires careful and detailed analysis across multiple facets. We have thoroughly reviewed the manuscript and find that the content presented is necessary to support our conclusions rigorously. Below, we provide point-by-point responses to the reviewer's comments and describe the corresponding changes made in the revised manuscript.

**This paper claims that it is better than previous work because it includes a more comprehensive treatment of the climate system. But it needs to make clear right at the beginning what it does not include. There is no treatment of UV changes (now possible with TUV incorporated in WACCM) and tropospheric chemistry does not include changes in photolysis. So all the conclusions have to be tempered by these omissions, and this has to be made clear in the abstract. The abstract focuses on SAI impacts, but that is not correct. This specific SAI scenario, with more forcing in the SH, produces direct effects there. So the results here are not general results for SAI, and that also needs to be made clear in the title and the abstract.**

We thank the reviewer for this important clarification. We realized that stating that the model does not include photolysis changes is incorrect. The photolysis rates are calculated using lookup tables that take into account the overhead ozone column and clouds (Emmons et al., 2020; Kinnison et al., 2017). However, this approach does not include the direct effects on actinic flux from aerosol scattering and absorption. In response, we have revised the abstract to acknowledge the limitations of our study more explicitly. We also emphasize that our analysis is specific to the ARISE-SA-1.5 scenario, which applies stronger forcing in the Southern Hemisphere.

We would also like to point out that even though a full interactive TUV scheme is now being incorporated into WACCM (as opposed to look-up table approach), it has only just been released and was not available at the time these CESM2(WACCM) simulations were carried out. Hence, if one wanted to use the new version, one would need to re-run both the 10-member ensemble of the control SSP2-4.5 simulations and all the SAI simulations, which is beyond the scope of the current study.

**It is certainly not correct to present results in the abstract to three significant figures without error bars, particularly since 10-member ensembles were used here.**

In the revised abstract, we now report results to two significant figures rather than three. In addition, we have included the ensemble range of PM2.5 and ozone-related mortality estimates to better reflect the uncertainty across the 10-member ensemble.

**In the title → "Small" → as compared to what?**

We appreciate the reviewer's request for clarification. By "small," we mean the additional air quality-related mortality impacts of SAI relative to those projected under the baseline SSP2-4.5 scenario. As shown in the abstract, the percentage changes in air-quality–related mortality under SAI are comparable to the magnitude of the changes projected under SSP2-4.5 alone.

**L 58: "comprehensive" → NO. It does not account for changes in UV due to ozone depletion, and its effects on ozone chemistry in the stratosphere or troposphere.**

"Comprehensive" is a term often used in connection with WACCM (including WACCM page: https://www2.acom.ucar.edu/gcm/waccm). We also would like to point out that the reviewer is mistaken, and that the current photolysis does take into account the overhead ozone column and, as such, can represent changes in UV from ozone depletion. We have therefore removed phrasing that described the model as "comprehensive" throughout the manuscript.

**L 74: How well does this model simulate PM2.5? Ozone? Please show evaluations of the control simulations as compared to observations. If not well, this model should not be used.**

We appreciate the reviewers' request to demonstrate how well CESM2-WACCM6 simulates PM2.5 and ozone in the control climate. Our simulations use CESM2–WACCM6 with the MOZART‑T1 tropospheric chemistry scheme and the MAM4 aerosol module, which have been extensively documented and evaluated in prior work. Gettlemen et al. (2019) showed that WACCM6 can reproduce observed climatology of trace constituents, in particular ozone, in the middle atmosphere. They concluded that it was capable of reproducing the evolution of ozone in the 20th and 21st centuries. Tilmes et al. (2019), Emmons et al. (2020) and Schwantes et al. (2020) further describe and evaluate the new chemistry mechanisms in CESM2(WACCM6) using both previous model versions and observations such as the NASA ATom aircraft mission, surface ozone data from the Tropospheric Ozone Assessment Report (TOAR) and carbon monoxide from the Measurements of Pollution in The Troposphere (MOPITT), finding good agreement with ozonesonde data and seasonality of surface ozone, while finding some spatial biases in some specific regions. Furthermore, Griffiths et al. (2020) benchmarked CMIP6 models, including WACCM, against observed tropospheric ozone distributions and found overall agreement in the spatial and seasonal variability. For aerosols and PM2.5, Hancock et al. (2023) used WACCM to study air quality over India and found that while the model underestimates PM2.5 in some regions due to missing secondary species (i.e., ammonium), the model

reproduces the observed spatial patterns and long-term trends. Taken together, these evaluations demonstrate that CESM2–WACCM6 reproduces the global and regional distributions of ozone and aerosol species with skill comparable to or better than previous CESM versions, and within the performance range of current state-of-the-art chemistry–climate models.

In response, we now state more clearly in the methods and conclusion sections the explicit caveats of WACCM6 while also emphasizing that our conclusions rely on scenario differences rather than absolute concentrations.

**L 106: "For PM2.5" → Doesn't it depend on the chemistry of the particles and not just their concentration?**

The toxicity of PM2.5 can vary depending on its chemical composition and source, and there is active research on quantifying such differences (Lippmann et al., 2013; Stanek et al., 2011). In this study, we follow the Global Burden of Diseases (GBD) framework, which applies a concentration–response relationship for PM2.5 exposure that does not distinguish by composition. While this represents a simplification, it enables consistency with widely used health impact assessments.

**L 118: What are each of these physically?  What are the units?**

The parameters θ, α, μ, and *v* are not physical quantities with units, but rather empirical coefficients of the Global Exposure Mortality Model (GEMM) function that defines the concentration–response relationship between PM2.5 exposure and relative risk of mortality. These parameters are estimated by fitting the IER function to epidemiological data for each cause of death and age group, and they are dimensionless. Their role is to shape the curve of the exposure–response relationship (e.g., slope, curvature, and inflection point) rather than representing a physical process. Table 1 serves as a reference for the specific parameter values.

**Equation 2 has multiple variables that are not explained.  What do each of them mean? What are the units?  And what is the science behind this equation?  Furthermore, where does the equation come from?  What is the reference?**

We thank the reviewer for pointing this out. In the revised manuscript, we have added explicit descriptions for all variables in Eq. 2, including units (e.g., PM2.5 concentration and relative risk (RR).  The scientific basis for equation 1 is that it follows the standard health impact function commonly used in air pollution epidemiology, which relates changes in pollutant concentrations to attributable health impacts. This formulation is consistent with the methodology presented in the 2015 EPA BenMAP User's Manual (US EPA), which we now cite directly in the text.

**How can OSMDA8 (the highest daily 8-hour average ozone concentration during the ozone season) be important for mortality?  Shouldn't the impact of ozone on mortality be**

**the amount of ozone times the exposure?  What if there are many days in a season with a little less ozone, and hence a lower OSMDA8, and in a different season only one day with a high OSMDA8 value and all the other days very low?  Wouldn't the first case be worse for health?  Please explain why the metric you are using makes sense.**

OSMDA8 is a common ozone metric used for human health in recent studies, such as in the Global Burden of Disease, an international effort, as well as many other studies (Malashock et al., 2022; Murray et al., 2024;  Sun et al., 2024). In order to match the methods used in these newer studies, we have decided to use it in this study as well.

OSMDA8 is not the highest daily 8-hour average ozone concentration during the ozone season, but rather the *highest 6-month rolling average* daily 8-hour average ozone concentration. The text has been revised to reflect this. This means that the OSMDA8 reflects the highest average MDA8 over a 6-month period, and is a good metric to use for long-term exposure to elevated ozone levels. The typical ozone season is from March to August in the Northern Hemisphere, and from September to February in the Southern Hemisphere, corresponding to their respective Spring and Summer months, but the 'ozone season' paradigm breaks down in the Tropics. The method of checking, for each grid box, where the 6-month rolling average MDA8 is maximized, allows for a better reflection of ozone exposure risk in the Tropics than just a blanket 'March to August North of the Equator and 'September to March' South of the Equator.

**Also, how much would ozone exposure affect mortality as a function of time over a person's lifetime?  Does it matter at what age they are exposed?**

Ozone exposure, similar to PM2.5, would likely have different RR values for different age groups. However, due to the lack of data on age-specific RR values, the Global Burden of Disease uses a single value for all adults (Murray et al., 2020). This assumption of a constant RR value with age has since been used in other studies as well (Malashock et al., 2022; Murray et al., 2024;  Sun et al., 2024). However, while the RR (i.e., the concentration–response function linking pollutant exposure to mortality risk)) is constant across age groups in our study, the baseline mortality rates (BMR) we apply are age-specific. For ozone, BMR refers specifically to mortality from respiratory and cardiovascular diseases, which determines how the exposure–response relationship translates into actual deaths. BMR for respiratory and cardiovascular diseases are much higher for older adults compared to younger adults. Thus, for the same RR value, premature deaths from respiratory and cardiovascular diseases due to ozone exposure are still higher for older populations.

**Lines 148-150.  You can't just choose to ignore uncertainty that you know about.  This will give you the wrong answers.  This just reinforces that the numbers in the abstract to 3 significant figures and no error bars can't possibly be correct.**

We thank the reviewer for this important point. Our analysis is designed to assess relative differences in mortality outcomes between scenarios using a consistent set of SSP2-4.5 inputs, rather than provide absolute projections with full quantified uncertainty bounds. We agree that

reporting numbers to three significant figures without error ranges overstated the precision of our results. In response, we have rounded all mortality estimates to two significant figures in the abstract.

**Fig. 3 has multiple issues:**
**- You have to use the same scales for all the panels in each row, like you did for Fig. 1, so that they can be compared.  Otherwise the same color means different things in each panel.**
**- The shading in Figs. 2-3 is hard to make out, as only two colors are used, and the boundaries between the different values are not clear.  Use distinct different colors.**
**- Needs stippling like in Fig. 1 to indicate which results are significant.**
**- Is it height above sea level?  How can you have values under the ice in Antarctica?**
**- Mark the latitude in more increments, and use natural ones, every 15 or 30 degrees.**
**- The font in the figures is too small to see.**
**- You plot water concentration, but is it water vapor or total water, including liquid and solid?  If water vapor, you have to use normal meteorological units of mixing ratio or absolute humidity.  And you show large changes in the Tropics, but the ITCZ has a large seasonal cycle and spatial variations.  Showing zonal-mean annual-mean values obscures much of the signal.**

We thank the reviewer for these detailed suggestions. Figure 3 has been revised to address the concerns raise: i) all panels within each row now share the same colorbar scale, enabling direct comparison, ii) stippling has been added, consistent with Fig. 2, to indicate regions that are not statistically significant across ensemble members, and iii) latitude increments are now shown at 30 degree intervals, and font size has been increased for readability. Regarding the plotted variable in Fig. 3j, k, and l, the figure shows the percentage change in water vapor concentration.

We agree that the ITCZ exhibits a strong seasonal cycle and spatial variability, and acknowledge that zonal-mean, annual-mean values inevitably obscure aspects of this signal. Because our health impact analysis is based on annual, population-weighted concentrations of OSMDA8 and PM2.5, the zonal-mean changes presented here are the most relevant for our study design. To provide additional context, we now include seasonal zonal-mean figures of water vapor mixing ratio (% changes) in the Supplementary Information to show how seasonal variations compare to the annual-mean changes, while keeping the main analysis focused on the health outcomes.

**H2O levels [What does it mean? Is it just water vapor?] Why would surface cooling produce this, in the ignorance of other factors, including circulation changes? —> "This reduces chemical ozone loss in the free-troposphere, as indicated by an increased net photochemical ozone production (Fig. 3c)."**

By 'H2O levels', we are referring to water vapor concentrations. Surface cooling would reduce ambient water vapor concentrations generally due to the Clausius-Clapeyron relationship. While

there could be regional changes in water-related quantities such as precipitation and soil moisture, the relationship between ambient water vapor concentrations and surface temperature are robust and identified as a mechanism that influences tropospheric chemistry under SAI (see for instance, Visioni et al., 2017).

**Explain how this happens: "This reduces chemical ozone loss in the free-troposphere, as indicated by an increase in net photochemical ozone production". What does water have to do with it?**

In daytime conditions, a primary route of ozone loss is via gas-phase photochemistry. Ozone ($O_3$) can undergo photodissociation reactions to generate energetic oxygen radicals ($O(^1D)$), which in turn can react with water vapor to generate hydroxyl radicals (OH).

$O_3 + h\nu \rightarrow O_2 + O(^1D)$
$O(^1D) + H_2O \rightarrow 2OH$

In unpolluted regions (aka regions with low nitrogen oxide ($NOx = NO + NO_2$) concentrations, such as the remote troposphere), further reactions between $O_3$ and OH can lead to catalytic $O_3$ losses:

$OH + O_3 \rightarrow HO_2 + O_2$
$HO_2 + O_3 \rightarrow OH + 2O_2$
Net: $2O_3 \rightarrow 3O_2$

The most recent IPCC AR6 report (Szopa et al., 2021) states with high confidence a positive relationship between increasing surface temperatures and increased $O_3$ loss via increasing water vapor concentrations (which, in turn, occurs due to the Clausius-Clapeyron relationship) over unpolluted areas. The reverse is also true: cooling reduces water vapor, weakens this pathway, and thereby reduces chemical ozone loss (Visioni et al., 2017; Bednarz et al., 2023). This appears as an increase in net photochemical ozone production in our simulations.

**L 186: "modifications in photolysis rates due to changes in stratospheric radiation…" What does this mean? In the stratosphere only? What wavelength radiation? Is it temperature dependent reactions? And you ignore changes in UV.**

We thank the reviewer for pointing out this contradiction. We have now rephrased this to read: "modifications in photolysis rates due to changes in actinic flux from changes in the overhead ozone column and aerosol absorption and scattering"

As mentioned, photolysis rates are included in the model but they are pre-calculated using lookup tables that take into account overhead ozone column and clouds  However, there are limitations with this approach in that they do not include changes caused by aerosol absorption and scattering, which we have now included in the text.

In addition, we have performed preliminary offline analyses of surface UV radiation differences between ARISE-SAI-1.5 and SSP2-4.5 using the Tropospheric Ultraviolet and Visible (TUV) model developed at NCAR to explore TUV changes in response to aerosol scattering and absorption. These results, now included in the Supplementary Information, indicate small changes but highlight a promising direction for further investigation.

**Figure A4: Text is much too tiny to read. Make the panels much bigger and use fewer per row. Since the color bar is the same for all the panels, get rid of the small ones and just use one large one. And what are the ////? It looks like Fig. A4 was done with GrADS, and it looks much better than the others, with better labeling of the axes and distinct colors for the shading. But why is there no white box behind each number in the contour labels, so they can be more easily read?**

Figure A4 and other figures in the Appendix are more appropriate for the supplemental materials and have been moved to the supplement. We have also replotted Figure A4 taking the reviewer suggestions into account.

**Figure 4: It is really hard to compare the two columns, as they need to be plotted with the same color scale. But it looks like the values in the left column for some countries like India and larger than the standard deviation. So why are they indicated as being significantly different from zero?**

Figures 4 and 6 have been replotted with the same color scale. The color scales now show that the values on the left column are larger than the standard deviation. The stippling is added over countries where the country's mean change is not statistically significant across ensemble members at 95% confidence interval.

**The paper uses "notably" randomly. These should all be deleted. Every sentence should be noted or it should not be in the paper.**

The use of "notably" and derivations of the word have been removed from the text.

**The paper references Fig. 7 before Figs. 5 and 6. This is confusing. Figures have to appear in numerical order in a paper.**

References to Fig. 7 in the first paragraph of Section 3.2 have been removed.

**Bibliography**

Adam, O., Bischoff, T., & Schneider, T. (2016). Seasonal and interannual variations of the energy flux equator and ITCZ. Part I: Zonally averaged ITCZ position. Journal of Climate, 29(9), 3219-3230.

Bednarz, E. M., Visioni, D., Kravitz, B., Jones, A., Haywood, J. M., Richter, J., ... & Braesicke, P. (2023). Climate response to off-equatorial stratospheric sulfur injections in three Earth system models–Part 2: Stratospheric and free-tropospheric response. Atmospheric Chemistry and Physics, 23(1), 687-709.

Emmons, L. K., et al. (2020). The Chemistry Mechanism in the Community Earth System Model Version 2 (CESM2). Journal of Advances in Modeling Earth Systems, 12(4), e2019MS001882. https://doi.org/10.1029/2019MS001882

Gettelman, A., Mills, M. J., Kinnison, D. E., Garcia, R. R., Smith, A. K., Marsh, D. R., ... & Randel, W. J. (2019). The whole atmosphere community climate model version 6 (WACCM6). Journal of Geophysical Research: Atmospheres, 124(23), 12380-12403.

Griffiths, P. T., Murray, L. T., Zeng, G., Shin, Y. M., Abraham, N. L., Archibald, A. T., ... & Zanis, P. (2021). Tropospheric ozone in CMIP6 simulations. Atmospheric Chemistry and Physics, 21(5), 4187-4218.

Hancock, S., Fiore, A. M., Westervelt, D. M., Correa, G., Lamarque, J. F., Venkataraman, C., & Sharma, A. (2023). Changing PM2. 5 and related meteorology over India from 1950–2014: a new perspective from a chemistry-climate model ensemble. Environmental Research: Climate, 2(1), 015003.

Lippmann, M., Chen, L. C., Gordon, T., Ito, K., & Thurston, G. D. (2013). National Particle Component Toxicity (NPACT) Initiative: integrated epidemiologic and toxicologic studies of the health effects of particulate matter components. Research Report (Health Effects Institute), (177), 5-13.

Malashock, D. A., Delang, M. N., Becker, J. S., Serre, M. L., West, J. J., Chang, K. L., ... & Anenberg, S. C. (2022). Global trends in ozone concentration and attributable mortality for urban, peri-urban, and rural areas between 2000 and 2019: a modelling study. The Lancet Planetary Health, 6(12), e958-e967.

Murray, C. J., Aravkin, A. Y., Zheng, P., Abbafati, C., Abbas, K. M., Abbasi-Kangevari, M., ... & Borzouei, S. (2020). Global burden of 87 risk factors in 204 countries and territories, 1990–2019: a systematic analysis for the Global Burden of Disease Study 2019. The lancet, 396(10258), 1223-1249.

Murray, C. J. (2024). Findings from the global burden of disease study 2021. The Lancet, 403(10440), 2259-2262.

Schwantes, R. H., Emmons, L. K., Orlando, J. J., Barth, M. C., Tyndall, G. S., Hall, S. R., ... & Bui, T. P. V. (2020). Comprehensive isoprene and terpene gas-phase chemistry improves simulated surface ozone in the southeastern US. Atmospheric Chemistry and Physics, 20(6), 3739-3776.

Stanek, L. W., Sacks, J. D., Dutton, S. J., & Dubois, J. J. B. (2011). Attributing health effects to apportioned components and sources of particulate matter: an evaluation of collective results. Atmospheric Environment, 45(32), 5655-5663.

Sun, H. Z., Van Daalen, K. R., Morawska, L., Guillas, S., Giorio, C., Di, Q., ... & Archibald, A. T. (2024). An estimate of global cardiovascular mortality burden attributable to ambient ozone exposure reveals urban-rural environmental injustice. One Earth, 7(10), 1803-1819.

Szopa, S., Naik, V., Adhikary, B., Artaxo, P., Berntsen, T., Collins, W. D., ... & Zanis, P. (2021). Short-Lived Climate Forcers (Chapter 6).

Tilmes, S., Hodzic, A., Emmons, L. K., Mills, M. J., Gettelman, A., Kinnison, D. E., ... & Liu, X. (2019). Climate forcing and trends of organic aerosols in the Community Earth System Model (CESM2). Journal of Advances in Modeling Earth Systems, 11(12), 4323-4351.

Visioni, D., Pitari, G., Aquila, V., Tilmes, S., Cionni, I., Di Genova, G., & Mancini, E. (2017). Sulfate geoengineering impact on methane transport and lifetime: results from the Geoengineering Model Intercomparison Project (GeoMIP). Atmospheric Chemistry and Physics, 17(18), 11209-11226.

US Environmental Protection Agency. (2018). Environmental Benefits Mapping and Analysis Program: Community Edition (BenMAP‑CE) User Manual and Appendices.

---

## Author Comment (AC2)

Reviewer comments are in **bold** and the authors' responses are in blue.

We thank the reviewer for their thoughtful and constructive evaluation of our manuscript. The central concern raised on whether CESM2-WACCM6 is an appropriate model for assessing air quality-related health impacts of SAI is an important one, and we have worked to address it directly. In the revised manuscript, we have clarified the limitations of CESM2-WACCM6 for air quality applications, drawing on previous work evaluating the model and highlighting where gaps remain. We have removed wording that overstated the comprehensiveness of the model, added explicit caveats in the abstract and discussion regarding photolysis and missing aerosol species, and reframed our conclusions. We have also expanded our interpretation of the conclusions in relation to the "climate penalty" literature, and added supplemental analysis to provide further context for some of our results. Below, we provide point-by-point responses to the reviewer's comments and describe the corresponding changes made in the revised manuscript.

**Major comments**

**The two most significant concerns are related, and boil down to the question of whether the model being used is appropriate for the task at hand. On the one hand, the authors make a good case that the fully interactive nature of the CESM2-WACCM6 simulation means that it can capture key meteorological responses to SAI, which are likely to be significant to the air quality response. Since these responses were often either neglected entirely or crudely parameterized in previous studies, directly simulating the interactions of changing meteorology with air quality is a valuable advance. However, the first question is whether the chosen model is appropriate for simulations of air quality. I am aware of almost no studies which have used WACCM for air quality modelling, beyond one study which is cited by the authors and which was itself an intercomparison of CMIP6 models. There seem to be several modelling choices in WACCM which, while sensible for a model of whole-atmosphere climate responses, might compromise its representation of air quality responses. For example Hancock et al. (2023) indicate that WACCM does not include any representation of ammonium or nitrate aerosols, but these are standard in models such as CMAQ which are dedicated to air quality – and such aerosols can be dominant in understanding air quality responses to climate change (see e.g. Nolte et al. (2018)). Indeed it appears that a recent paper in ACP which used WACCM for the boundary conditions in an air quality simulation specifically chose to use WRF for the regional analysis, in part because it includes air quality-relevant aerosol chemistry lacking in WACCM (Clayton et al., 2024). I would strongly recommend that the authors perform a detailed evaluation of a) WACCM's ability to represent baseline air quality in the present day, b) WACCM's ability to reproduce already-understood effects of climate change on air quality, and c) the likely gaps in WACCM's representation of processes and species which are important to air quality, beyond the question of tropospheric photolysis below.**

We agree that the choice of model is central to the credibility of our conclusions, and we appreciate the opportunity to clarify both the strengths and limitations of CESM2-WACCM6 for air quality analysis.

First, regarding model suitability, we chose to analyse the CESM2-WACCM6 ARISE-SAI simulation output because its fully interactive chemistry–climate framework allows us to capture the coupled response of meteorology, radiation, and atmospheric composition under SAI. ARISE-SAI simulations have already been widely applied to study climate impacts of SAI, and we believe expanding its use to study air quality is a necessary next step. Although not without limitations, CESM's integration of climate, radiation, chemistry, and aerosols provides a unique opportunity to evaluate potential on-the-ground impacts of SAI. This coupling capability is essential, as air quality responses to SAI are strongly mediated by dynamical and chemical feedbacks that cannot be represented in offline or regionally constrained models. While not perfect, our analysis offers a valuable foundation for more comprehensive assessments of air quality and SAI. Moreover, CESM2-WACCM6 has previously been evaluated against observations of ozone, aerosols, and precursors. Emmons et al. (2020) provide a systematic assessment of CAM6-chem and WACCM6, showing that the model captures large-scale distributions of tropospheric ozone and key pollutants. Similarly, Gettelman et al. (2019) evaluate WACCM6's baseline climatology and variability, finding that its representation of ozone, aerosols, and chemistry is consistent with other state-of-the-art Earth system models. Griffiths et al. (2021) benchmark CMIP6 models (including WACCM) against observed tropospheric ozone distributions, demonstrating broad agreement in spatial and seasonal variability. Additional studies highlight the implementation of new chemistry mechanisms: Tilmes et al. (2019), Emmons et al. (2019), and Schwantes et al. (2020) evaluated CESM2(WACCM6) against both earlier model versions and multiple observational datasets—including NASA's ATom aircraft campaign, TOAR surface ozone, and MOPITT carbon monoxide—reporting good agreement with ozonesondes and seasonal ozone cycles, though with some regional spatial biases. For aerosols, Hancock et al. (2023) used CESM2-WACCM6 to analyze PM2.5 over India, finding that while the model underestimates concentrations in some regions due to missing species, it reproduces observed spatial patterns and long-term trends.

Regarding WACCM's ability to reproduce effects of climate change on air quality, WACCM has also been applied to evaluate well-established climate–air quality interactions. Fiore et al. (2022) demonstrate that WACCM6 reproduces observed large-scale tropospheric ozone changes in response to climate variability and anthropogenic forcing. Griffiths et al. (2021) similarly show that WACCM captures the long-term evolution of tropospheric ozone, consistent with our understanding of emissions and climate drivers.

At the same time, important limitations must be acknowledged. Regarding Hancock et al. (2023), we appreciate the reviewer raising this point. Indeed, CESM2-WACCM6 does not include explicit ammonium or nitrate aerosol chemistry, and these species can be important contributors to fine particulate matter in certain regions. Hancock et al. (2023) evaluated the performance of WACCM6 using observations of monthly PM2.5 over India and found that the model underestimates PM2.5 for certain seasons and cities due to the omission of coarse

particles, such as nitrate and ammonium, which are important components of PM2.5 in India. Ren et al. (2024) show that many CMIP6 models, including WACCM, underestimate PM2.5 burdens globally due to this omission.

Despite these limitations, Hancock et al. (2023) evaluated the spatial pattern and trends of PM2.5 and meteorological variables and concluded that air pollutant emissions, rather than climate variability, play a dominant role in poor air quality in India. In our analysis, air quality impacts are based on changes in PM2.5 and ozone. PM2.5 in WACCM is composed of six species (sulfate, organic carbon, black carbon, sea salt, and dust), so while ammonium is not included, the model still captures the major contributors to global PM2.5, and it is unlikely that sulfate-related changes would greatly affect ammonia contributions to PM2.5, making our conclusions robust to this shortcoming. That said, we recognize that ammonium and nitrate could add to the PM2.5 burden, especially in ammonia-rich regions, and this omission may lead us to underestimate pollution-related health impacts. We will make these limitations clearer in the revised manuscript (L 86).

**The second concern is related. Specifically, the fact that WACCM uses fixed tropospheric photolysis rates is a significant shortcoming in a study which seeks to understand the atmospheric composition implications of stratospheric aerosol injection. This is a difficult issue to rectify, and I am glad to see that the authors have at least acknowledged this challenge. However, previous studies (e.g. Xia et al., 2017) did include this response and discussed at length the potential for tropospheric UV changes to be significant in understanding the tropospheric ozone response – and thus the air quality response. The authors themselves argue that tropospheric photochemistry is the dominant factor in NH surface ozone change (line 219). Ideally, an analysis such as that by Clayton et al. (2024) in which WACCM outputs are used as boundary conditions to a more air quality-focused model may be a way to resolve these issues, and I would recommend that the authors seriously consider if there is a way that they could perform a more comprehensive simulation of tropospheric chemistry using their existing data - recognizing that this would require a great deal of additional work but would also resolve what I perceive as being a major gap in the work.**

We realized that stating that the model does not include photolysis changes is incorrect. The model includes photolysis rates that are calculated using lookup tables accounting for overhead ozone column and clouds. However, this approach of calculating the photolysis rates does not include the direct radiative effects of dynamic aerosol distributions. In an attempt to address this gap, we conducted offline calculations with the Trospheric Ultraviolet and Visible (TUV) model, which indicate that surface UV changes under SAI are small and broadly consistent with past studies (Bardeen et al., 2021).

Our analysis focuses on the health impacts of particulate matter (PM2.5), which is the primary driver of air pollution–related mortality worldwide. While we acknowledge that our framework likely underestimates the role of tropospheric photochemistry in shaping ozone changes, the main conclusions regarding PM2.5 impacts remain robust. We have highlighted this limitation in

the text and interpreted the ozone-related results with appropriate caution (L 380). We view our study as a first step in quantifying global-scale mortality implications of SAI and hope that future work can build upon it by explicitly incorporating variable tropospheric photolysis rates and extending the analysis to UV-driven health outcomes (e.g., skin cancer, cataracts) that lie outside the scope of the present paper.

**Notwithstanding such an expansion, these are sufficiently significant deficiencies that I believe they need to be much more strongly highlighted. I would recommend that the abstract explicitly state that changes in tropospheric photolysis are not considered, and that statements that this is the first study to use "comprehensive" stratospheric and tropospheric chemistry (e.g. line 29 and 58) be removed. While I absolutely believe that this study can provide a valuable contribution to our understanding of the impacts of SAI on the environment, I would argue that it needs to be placed in the correct context (and thus allow subsequent studies to fill the remaining knowledge gaps).**

We have revised the manuscript to remove the phrasing of the model as "comprehensive" and avoid suggesting that this is the first study with fully comprehensive stratospheric and tropospheric chemistry. In the abstract, we have added explicit caveats about photolysis/UV treatment and clarified that the results are scenario-specific, not general for all SAI.

**Independent of these concerns, I was struck by one of the conclusions drawn (and which is highlighted in the abstract). The authors argue that internal variability is key, on the basis that they find significant differences across ensemble members. This aligns climate intervention effects on air quality with the well established effects of climate change on air quality (e.g. Fiore et al., 2015) where noise in the meteorological response can be greater than the change in exposure to pollutants resulting from SAI. It would have been useful to discuss how the projected effects of SAI on air quality compare to the air quality "penalty" projected for climate change, given that there is a robust literature discussing not only this question but also specifically the problem of how to deal with internal variability in such projections. The lack of such a discussion is a notable absence, and leaves the paper somewhat unmoored.**

We thank the reviewer for highlighting the importance of exploring the parallel between internal variability in climate intervention studies and the air quality "penalty" literature. We agree that this is an important contextual point, and we have added a new discussion at the end of the conclusion (see page 18, last paragraph) comparing our findings to prior work.

**The use of large ensembles is a good (if expensive) solution to this problem, but analysis of air quality interventions may also rely on representative meteorological years if it can be shown that the outcome would be the same as when using a large ensemble average (see e.g. Stewart et al. (2017) and Abel et al. (2018) for examples looking at air quality change in future conditions). Here it seems that internal variability is used to draw some conclusions which seem hard to justify; for example, on lines 291-293 it is claimed that "health impacts under SAI are not governed mainly by the magnitude of SO2 injected".**

**Certainly it is true that SAI alone is not going to become the dominant cause of air pollution under almost any scenario, and the comparison of ARISE-SAI-1.0 and 1.5 shows how important these other factors are - an important contribution. However the paper simultaneously argues that there is a robust surface ozone response relative to a scenario where the amount of SO2 injected is zero (SSP2-4.5), so presumably the magnitude of the injection is not entirely irrelevant. Is there evidence that a robust (if complex) difference in the effects of larger injection quantities would not emerge if using a larger ensemble, longer averaging period, and/or if other factors (eg surface-level emissions of air quality precursors) were held constant? I would suggest that the authors explore in more detail the degree to which their results might be improved by such approaches, not least because the data to do so appears to already exist (e.g. it should be straightforward to evaluate the degree to which a smaller ensemble would or would not have allowed the same conclusions to be drawn - which would be valuable information for those interested in performing future studies of atmospheric composition change under SAI).**

We thank the reviewer for raising this important point. In Figs. 7 and 8, we show that air pollution-related mortality does not increase monotonically with the magnitude of SO2 injected across ARISE-SAI-1.0 and ARISE-SAI-1.5. There is a robust surface ozone response, which likely arises from SO2 being injected primarily in the southern hemisphere during 2060-2069, that modulates the hemispheric temperature gradients. For surface ozone, the impact on mortality is a complex interplay between deposition, tropospheric and stratospheric changes due to chemistry and transport, and surface changes due to cooling. However, our results indicate that such an interplay results in a significant, but not magnitude-dependent, change. Furthermore, changes in surface ozone impact the spatial distribution of ozone-related mortality but not the global average.

In additional simulations following the G2-SAI-3DOF and G2-SAI-1DOF protocols (Visioni et al., 2024), where injections occur *without* any associated changes in tropospheric chemistry, we still find a robust surface ozone response relative to SSP2-4.5.

Regarding ensemble size, our analysis draws on two independent 10-member ensembles (ARISE-SAI-1.5 and ARISE-SAI-1.0). A 10-member ensemble is generally considered sufficient to separate forced responses from internal variability (Milinski et al., 2020; Wills et al., 2020), and indeed, our results show consistent outcomes across both ensembles. Furthermore, by analyzing 35-year time series (see figure below), we find that the overall mortality estimates do not exhibit a clear trend with increasing injection amounts, regardless of the temporal averaging resolution. Therefore, having a large ensemble size gives us more confidence in our results. However, we agree that using representative meteorological years or expanding the ensemble size further could provide additional insights, and we have revised the text to explicitly acknowledge this (last sentence of Section 3.2). To address this concern, we performed a subsampling analysis: drawing 10 random sets of 3 ensemble members each, we found that only a minority of subsamples exhibited a statistically significant linear relationship (p < 0.05) between injection rate and mortality, and even then, the correlation coefficients were weak. This

suggests that our central conclusion that mortality impacts do not scale in a simple linear fashion with SO2 injection remains robust, though further ensembles, longer time periods, and controlled precursor emissions experiments would be valuable in future work. Both figures have been added to the supplemental.

[Figure]

Global differences in air-pollution mortality between ARISE-SAI and SSP2-4.5 as a function of the total SO2 injection rate for the first 35 simulation years. Panels show (a) PM2.5-attributable mortality for ARISE-SAI-1.5, (b) ozone-attributable mortality for ARISE-SAI-1.5, (c) PM2.5 for ARISE-SAI-1.0, and (d) ozone for ARISE-SAI-1.0. Colored lines indicate individual ensemble members, while the thick black line represents the 10-member ensemble mean.

[Figure]

Global mortality differences between ARISE-SAI and SSP2-4.5 are shown as a function of the SO2 injection rate over the first 35 years of simulation. Panels depict (a) PM2.5-attributable mortality for ARISE-SAI-1.5, (b) ozone-attributable mortality for ARISE-SAI-1.5, (c) PM2.5 for ARISE-SAI-1.0, and (d) ozone for ARISE-SAI-1.0. Colored lines represent ordinary least-squares fits from randomly selected 3-member ensemble subsets (10 draws), plotted only where significant linear relationships are detected (p < 0.05). Slopes, r-values, and p-values are annotated in the legend.

**Minor comments**

**Some aspects of the air quality response which I had expected might be significant were seemingly not discussed. I would recommend discussing whether elements of the air quality response to SAI which have been significant for studies of the climate penalty – for example, changes in planetary boundary layer height, and the (highly model-dependent) lightning response – are playing a significant role in the calculated response. These factors are well described in the literature already cited and would be expected to be represented in an ESM (ostensibly one of the key novelties of this work), so providing a careful evaluation of how these factors translate to an SAI study would be valuable.**

We appreciate the reviewer's suggestion and agree that factors such as planetary boundary layer (PBL) height and lightning are important components of the broader air quality–climate

literature. However, the scope of the present study is intentionally focused on quantifying the air quality and associated health impacts of SAI, using CESM to evaluate the net surface-level changes in PM2.5 and ozone concentrations across large ensembles. Our goal here is not to provide a mechanistic attribution of every pathway by which SAI may influence surface air quality, but rather to assess the aggregate outcome of these multiple processes as represented within the model.

Elements such as PBL height and lightning response are indeed simulated in WACCM and therefore implicitly contribute to the overall modeled response. To address the reviewer's comments, we have included a plot of PBLH changes in the supplemental which show interesting changes in PBLH from SAI but a more detailed process-level analysis of each of these mechanisms is an important and valuable direction for future work. We frame our analysis around the ensemble-mean concentration and mortality responses, and we highlight where internal variability and policy-driven changes dominate the signal. By design, this allows us to place the air quality consequences of SAI in direct context with prior studies of the climate penalty and emissions controls, while keeping the focus on the net implications for surface air quality and health outcomes. We have clarified this point in the text and noted that more detailed process studies, which include explicit evaluation of changes in PBL dynamics and lightning, will be a valuable complement to our findings (L 385).

**While I understand why the authors have chosen not to estimate the health impacts of UV changes associated with SAI, I was surprised that no formal analysis was done at all of surface UV changes. The statement on line 372 – that a preliminary analysis indicated "very modest changes" – is unfortunately not much help, as the authors do not provide any metric of what they consider to be "modest" (or why). Quantifying (say) relative changes in projected population exposure to UV would help us to understand whether such changes need further study. Quantitative analysis of UV changes may also be useful in understanding the degree to which neglecting changes in tropospheric photolysis change may or may not be a minor oversight.**

We agree that quantifying surface UV changes provides useful context for understanding both health and chemical implications. For the preliminary analysis in question, we conducted an offline analysis of the UV changes using the Tropospheric Ultraviolet and Visible Radiation Model (TUV-X; https://github.com/NCAR/tuv-x). We calculated photolysis rate constants and surface UV changes under clear-sky conditions, comparing output from the two simulations. Our analysis indicates that percentage changes in surface UV are between -5.3 to -6.1% globally. For example, for JJA 2069 we find relative changes on the order of only a few percent between the SAI and SSP2-4.5 scenarios.

In response to this comment, we have added a supplemental figure (shown below) showing the spatial distribution of percentage changes in surface UV for JJA 2069. This figure illustrates that changes are small across nearly all regions. While this analysis confirms that UV changes are not a dominant driver of the air quality responses we focus on here, we agree that they remain relevant for future work, especially in the context of quantifying potential UV-related health

effects. The text has been revised to include a discussion of our preliminary findings from TUV and how it is consistent with previous work (L 391).

[Figure]

**2069 JJA UV Dose Rates**
**ARISE-SAI-1.5 minus SSP2-4.5**

a) UV-A, 315-400 nm: -6.1%

b) UV-B, 280-315 nm: -5.3%

c) UV-B*, 280-320 nm: -5.3%

% Difference

**Hancock et al. (2023) indicated that WACCM-based estimates of exposure to PM2.5 may overestimate the role of dust, due to inclusion of too-large particles in the PM2.5 metric.**

**Given that dust is the predominant factor in exposure under ARISE-SAI-1.5 for a significant fraction of the world (Figure 2), it would be useful to have more information on how the PM2.5 calculation was performed and whether the issue identified by Hancock et al. was addressed.**

The reviewer raises an important point. Our PM2.5 calculation follows the same setup described in Hancock et al. (2023), and it is therefore subject to the same caveats regarding the representation of dust, including the potential inclusion of overly large particles in the PM2.5 metric. We acknowledge that this may lead to some overestimation of dust contributions to total exposure. However, we emphasize that our analysis focuses on differences between the ARISE-SAI-1.5 and SSP2-4.5 scenarios, rather than the absolute magnitudes of exposure. Because the same definition of PM2.5 is applied consistently across both scenarios, any systematic bias in the representation of dust is expected to cancel out when examining the relative effects of SAI. For this reason, while the caution identified by Hancock et al. is relevant to the interpretation of the absolute dust burden, it does not materially affect the conclusions we draw about the differences attributable to SAI. We have included text in the manuscript to highlight this caveat.

**There are numerous grammatical errors (e.g. lines 221-222: "many of this conditions", "we deem important"; line 228: "These estimates and Fig. 4 show that the standard deviation of mortality estimates highlights the large spread in project PM2.5-related deaths"; Eq. 2 says the PM2.5 threshold is 2.4 (no units given), but Table 1 says 2.5 ppm - and Burnett et al (2018) say 2.4 ug/m3). I would recommend the authors take some time to go through the paper in depth and fix such errors before resubmitting.**

The text has been revised to address these grammatical errors.

**Bibliography**

Bardeen, C. G., Kinnison, D. E., Toon, O. B., Mills, M. J., Vitt, F., Xia, L., ... & Robock, A. (2021). Extreme ozone loss following nuclear war results in enhanced surface ultraviolet radiation. Journal of Geophysical Research: Atmospheres, 126(18), e2021JD035079.

Emmons, L. K., et al. (2020). The Chemistry Mechanism in the Community Earth System Model Version 2 (CESM2). Journal of Advances in Modeling Earth Systems, 12(4), e2019MS001882. https://doi.org/10.1029/2019MS001882

Fiore, A. M., Hancock, S. E., Lamarque, J. F., Correa, G. P., Chang, K. L., Ru, M., ... & Ziemke, J. R. (2022). Understanding recent tropospheric ozone trends in the context of large internal variability: a new perspective from chemistry-climate model ensembles. Environmental Research: Climate, 1(2), 025008.

Gettelman, A., Mills, M. J., Kinnison, D. E., Garcia, R. R., Smith, A. K., Marsh, D. R., ... & Randel, W. J. (2019). The whole atmosphere community climate model version 6 (WACCM6). Journal of Geophysical Research: Atmospheres, 124(23), 12380-12403.

Griffiths, P. T., Murray, L. T., Zeng, G., Shin, Y. M., Abraham, N. L., Archibald, A. T., ... & Zanis, P. (2021). Tropospheric ozone in CMIP6 simulations. Atmospheric Chemistry and Physics, 21(5), 4187-4218.

Hancock, S., Fiore, A. M., Westervelt, D. M., Correa, G., Lamarque, J. F., Venkataraman, C., & Sharma, A. (2023). Changing PM2. 5 and related meteorology over India from 1950–2014: a new perspective from a chemistry-climate model ensemble. Environmental Research: Climate, 2(1), 015003.

Milinski, S., Maher, N., & Olonscheck, D. (2020). How large does a large ensemble need to be?. Earth System Dynamics, 11(4), 885-901.

Nolte, Christopher G., et al. The potential effects of climate change on air quality across the conterminous US at 2030 under three Representative Concentration Pathways. Atmospheric Chemistry and Physics. 2018

Ren, F., Lin, J., Xu, C., Adeniran, J. A., Wang, J., Martin, R. V., ... & Takemura, T. (2024). Evaluation of CMIP6 model simulations of PM 2.5 and its components over China. Geoscientific Model Development, 17(12), 4821-4836.

Schwantes, R. H., Emmons, L. K., Orlando, J. J., Barth, M. C., Tyndall, G. S., Hall, S. R., ... & Bui, T. P. V. (2020). Comprehensive isoprene and terpene gas-phase chemistry improves simulated surface ozone in the southeastern US. Atmospheric Chemistry and Physics, 20(6), 3739-3776.

Tilmes, S., Hodzic, A., Emmons, L. K., Mills, M. J., Gettelman, A., Kinnison, D. E., ... & Liu, X. (2019). Climate forcing and trends of organic aerosols in the Community Earth System Model (CESM2). Journal of Advances in Modeling Earth Systems, 11(12), 4323-4351.

Wills, R. C., Sippel, S., & Barnes, E. A. (2020). Separating forced and unforced components of climate change: the utility of pattern recognition methods in large ensembles and observations. Variations, 18(2), 1-10.

---

## Author Comment (AC3)

Reviewer comments are in **bold** and the authors' responses are in blue.

We thank the reviewer for their evaluation of our manuscript and for raising several important concerns. We have taken these comments seriously and have made revisions throughout the paper. In particular, we have clarified the limitations of CESM-WACCM6, adjusted language that overstated the comprehensiveness of the chemistry representation, rounded and reformatted numerical results to avoid overprecision, and expanded the conclusion to provide more interpretation of the results. We appreciate the reviewer's concern regarding the length of the manuscript. While the overall message may appear conceptually straightforward, arriving at this conclusion requires careful and detailed analysis across multiple facets. We have thoroughly reviewed the manuscript and find that the content presented is necessary to support our conclusions rigorously. Below, we provide point-by-point responses to the reviewer's comments and describe the corresponding changes made in the revised manuscript.

**Major comments**

**A key issue with this paper is the use of CESM2-WACCM6 for the evaluation of health impacts of stratospheric aerosol injection (SAI). Previous studies have relied on models with shortcomings noted in the introduction, but unfortunately the shortcomings of the model used here are downplayed. Main issues are with the fixed photolysis rates and lack of ammonium and nitrate aerosols, essential to air quality assessment. Without these terms, the author's conclusions are, at best, incomplete.**

**These are not minor caveats, but fundamentally constrain the reliability of this study, something the authors should be upfront about. Claims of "comprehensive chemistry" should therefore also be removed.**

We thank the reviewer for raising this important point. We agree that CESM2-WACCM6 has limitations that need to be acknowledged when interpreting our results. In particular, 1) photolysis rates are calculated using lookup tables that account for overhead ozone column and clouds but that do not include direct effects of dynamic aerosol distributions, and 2) the MAM4 aerosol module does not represent ammonium and nitrate, which are important contributors to PM2.5 in many regions. These are important caveats, and we have revised the manuscript to emphasize them more clearly.

While "comprehensive" is a term often used in connection with WACCM (including on the WACCM model webpage: https://www2.acom.ucar.edu/gcm/waccm), we recognize that its use here could be misleading. We have therefore removed phrasing that described the model as "comprehensive" throughout the manuscript.

**The use and interpretation of the 10-member ensembles in this manuscript raises several problems. The paper stresses variability across ensemble members yet lacks a systemic metric for uncertainty analysis. Without this, the role of ensembles remains descriptive, rather than analytical.**

We appreciate the opportunity to clarify the interpretation of the 10-member ensembles. For geographical figures (Figs. 1-4 and 6), our focus is on the spatial patterns and whether simulated changes are robust relative to internal variability. In these cases, we use stippling based on a two-sided t-test across ensemble members at the 95% confidence level to indicate regions where results are not statistically significant. For Fig. 2, we are interested in identifying the dominant PM2.5 species for each geographical region, so we stipple over areas where fewer than 90% (9 out of 10) of the ensemble members agree on the dominant species. For Figs. 5, 7-9, we focus on summarizing the overall magnitudes and regional differences in mortality burdens. Here, error bars show ensemble spread, since these quantities are spatially averaged and the spread directly conveys the variability in the estimated mortality burden. Thus, uncertainty is presented in a manner appropriate to the scientific question addressed by each figure.

**Simultaneously, there is the issue of overprecision. Results are given such as "-149,397 to -177,296" (line 227) which is indefensible when variability and uncertainty are ignored, as the authors admit to in lines 148-150. Results should be rounded and expressed as mean +- standard deviation or 90% confidence interval. Not as exact integers or with unrealistic significance. Unfortunately, variability is highlighted when it dilutes signal but downplayed when the results look robust. Such inconsistency weakens the conclusions.**

We appreciate this observation and agree that reporting excessively precise integers can be misleading. In the revised manuscript, all mortality estimates have been rounded to avoid the appearance of overprecision. For uncertainty, we have chosen to present the ensemble spread (minimum-maximum across members) rather than a +- standard deviation of 90% confidence intervals. Our primary approach to uncertainty is to present the spread across ensemble members, which reflects the internal variability captured by the model. The following text has been added to make this clear:

"In the figure, this variability is quantified by the standard deviation across mortality estimates, while the text emphasizes the overall spread in the projections."

**Furthermore, the statement that "mortality impacts do not scale with SO2 injection" is unsupported. Only two scenarios are compared, over a relatively short time period. A more nuanced treatment would recognize that non-linearity is plausible but cannot be demonstrated here.**

We thank the reviewer for this comment. Our analysis is based on two distinct large ensembles (ARISE-SAI-1.5) and ARISE-SAI-1.0), which span a wide injection range– from near 0 up to ~20 Tg So2/yr– sufficient to encompass a range of cooling from present-day conditions to ~3 degrees C. Over the 50-year simulation period, these scenarios provide a reasonable basis for assessing differences in health outcomes under SAI as a function of injection rate. However, we have revised the manuscript to clarify that while our results suggest no scaling of mortality impacts of SO2 injection across these scenarios, a more systematic assessment of non-linear

responses would require additional scenarios beyond those available here (last sentence of Section 3.2).

**The scope of this work is narrow. While the paper claims that "this study focused on the air quality-related health impacts of SAI", only ozone and PM2.5 are considered. The abstract (and title) should reflect the scope. Unfortunately, the paper glosses over the significant regional increases in mortality (Figure 9), these results deserve more emphasis, as focusing on global aggregates risks misinterpretation of the overall findings.**

We thank the reviewer for raising this important point. PM2.5 and ozone are the two pollutants most commonly used to represent surface air quality in global health assessments (Pandley et al., 2019), as they account for the vast majority of air quality-related mortality (GBD (2019), WHO, 2016 & 2021)). Thus, by focusing on PM2.5 and ozone, our study focuses on air quality-related health impacts of SAI.

While many prior studies (e.g., Eastham et al., 2018 and Harding et al., 2024) have focused primarily on global aggregates, we sought to go further by including Section 3.3, which presents mortality burdens for each GBD super-region. This regional perspective highlights heterogeneity in outcomes, including regions where mortality increases under SAI, even though the paper does not explore each regional change in detail. To avoid misinterpretation, we have revised the abstract to explicitly note that our analysis is limited to ozone- and PM2.5-attributable mortality and that both global and regional changes are considered. We believe this provides sufficient context for readers without changing the overall framing of the paper. We have included figures showing the regional changes PM2.5 and ozone-related mortality from SAI in the Supplemental.

**The discussion largely restates results rather than interpreting them. The reader is left with little beyond "impacts are modest".**

We agree with the reviewer that the discussion should go beyond restating results. To address this, we have expanded the conclusion to interpret our findings in the broader context of climate change and air quality policy. Specifically, we highlight the impacts of SAI relative to internal variability and policy-driven improvements, and relate our results to the "climate penalty" literature. We now emphasize that white SAI alters the spatial distribution of ozone and PM2.5, the dominant determinant of future health outcomes remains the strength of air quality policies. This additional discussion clarifies that our main conclusion isn't simply that the impacts are modest, but they are modest relative to variability and policy effects (SSP2-4.5), underscoring the importance of emissions reductions for long-term air quality and health (see last paragraph of conclusion).

**Minor comments:**

**Figres are dense, inconsistently referenced, and hard to interpret. At worst they are misleading (e.g., different scales across panels in Figure 3). Figure 5(b) is never referred to in the text. Remove this from the paper or discuss the meaning in the main body.**

We thank the reviewer for this helpful feedback. Figure 3 has been revised so that all panels now show percent changes and each row has consistent percentage-change scales across the three panels. We have also reviewed and corrected all figure references to ensure consistency. In addition, Fig 5b is now explicitly discussed in the main text.

**Check the citations. E.g., line 123: WHO cited as "(Organization et al., 2021).**

The citation has been corrected.

**Line 16: I strongly recommend against using uncommon words like "ameliorate".**

The word "ameliorate" has been replaced with the word "offset"

**Line 47&48: awkward use of "they" to refer to Harding et al., I suggest referring to the studies instead of the authors when critiquing methods used.**

In Line 47&48, the pronoun "they" refers to solar dimming simulations rather than to Harding et al. To avoid ambiguity, we have revised the sentence to explicitly state "solar dimming approaches" instead of "they.

**Line 54: "the the"**

The extra "the" has been removed from the text.

**Line 158: "these three-way comparison".**

The text has been revised to refer to "this three-way comparison" rather than "these three-way comparison"

**Line 184: this is phrased rather unprofessionally: I suggest replacing ", as in" with: "i.e.,".**

The text has been revised accordingly.

**Bibliography**

GBD 2019 Risk Factors Collaborators. Global burden of 87 risk factors in 204 countries and territories, 19902019: a systematic analysis for the Global Burden of Disease Study 2019 Lancet 2020; 396: 1223–49.

Pandey, A., Brauer, M., Cropper, M. L., Balakrishnan, K., Mathur, P., Dey, S., ... & Dandona, L. (2021). Health and economic impact of air pollution in the states of India: the Global Burden of Disease Study 2019. The Lancet Planetary Health, 5(1), e25-e38.

World Health Organization. (2016). Ambient air pollution: A global assessment of exposure and burden of disease. Clean Air Journal, 26(2), 6-6.

World Health Organization. (2021). WHO global air quality guidelines: particulate matter (PM2. 5 and PM10), ozone, nitrogen dioxide, sulfur dioxide and carbon monoxide. World Health Organization.

---

## Referee Report (RR1)

**Abstract:**
The addition of average and ensemble range of pollution-related mortality is good; the same applies to the reduction in ozone-related deaths. Adding the spread and explanation is good. The disclaimer in lines 15 – 18 are also good.

**Introduction:**
Line 28: "release of precursors such as sulfur dioxide" – precursors to what? Admittedly, readers are not likely to be confused by this, but the sentence should be made precise.

Line 77: the additional explanation of shortcomings is a welcome addition.

**Model description:**
Line 93: CESM2(WACCM6) is already defined (and used) as an abbreviation earlier in the text (lines 39-40, albeit without version).

Line 109: Suggest moving "(Wen et al, 2023; Wei and Tahrin, 2024)" to end of sentence in line 110, considering this second part contains the findings of said papers.

Line 131: "with the aim of maintain[ing]"

**Results:**
Prior to "In Fig. 1, …" I would highly recommend the authors to add a topic sentence, with the main conclusion of this paragraph. This is completely optional, but paragraph 3.1 currently reads like a summation of figures first and foremost, and less like an active interpretation of these findings. A topic sentence with the main findings before diving into the figures would be a welcome addition here.

The results section often mentions ensemble spread but does not clearly identify which results are robust across the ensemble and which are dominated by variability. For example, the hemispheric ozone asymmetry appears as a strong and consistent feature, whereas several PM2.5 patterns do not. Clarifying this distinction will help readers assess confidence in each result.

Related to this, many maps contain extensive stippled regions indicating non-significant differences. In some cases, the text still describes spatial patterns along these regions. It would be helpful to explicitly state when results are not statistically significant rather than implying interpretability from noisy patterns.

Line 216: "ITCZ" is defined, but not used again, so this (ITCZ) can be omitted.

Figure 3.: I suggest being consistent and writing out "percent" all the time, or even use "relative", instead of switching between "percent" and "%".

---

## Author Response (AR2)

**Response to Reviewer #1**

Reviewer comments are in **bold** and the authors' responses are in blue.

We thank the reviewer for their thorough and constructive evaluation of our manuscript. In response, we have removed unsupported assertions about nitrate and dust biases canceling between scenarios, clarified that nitrate–ammonium interactions are not represented in CESM2(WACCM6), and explicitly noted that WACCM's dust overestimation may inflate the prominence of dust in Fig. 2. We have also added the previously missing caveat in Sec. 2.1 and revised the title to avoid ambiguity around the term "deposition."

**First, the authors argue that the lack of nitrate aerosol is unlikely to cause a systematic bias because they are comparing ARISE-SAI-1.5 and SSP2-4.5 (lines 111-113). I do not understand this argument. Surface-level sulfate aerosol competes with nitrate aerosol, and can indeed displace it; furthermore nitrate aerosol/gas partitioning will change in response to near-surface temperature whereas near-surface sulfate partitioning is insensitive to temperature. This feels particularly significant given that SAI is expected to change, among other things, near-surface temperature. The lack of nitrate or ammonium aerosols in WACCM means that these interactions are not captured, and it is very difficult to say what their implications would be; consider, for example, work such as Tai et al. (2012) which suggests that nitrate specifically exhibits a different trend with climate change than other constituents. The statement that nitrate is "standard in some regional air quality models" (line 105), implying that the absence of nitrate in CESM2-WACCM for an air quality assessment is normal, also risks being somewhat misleading considering that CESM2-WACCM was the only model out of 6 global Earth system models in a recent intercomparison which lacked an explicit representation of nitrate gas/aerosol partitioning (He et al., 2025). I would recommend removing assertions regarding the size of the effect entirely and simply stating that WACCM does not include this interaction, as there is no clear evidence that the associated errors are negligible.**

We thank the reviewer for this important clarification. We agree that nitrate–ammonium thermodynamics and nitrate–sulfate competition can significantly affect PM2.5 levels and that the implications of omitting nitrate aerosol cannot be assumed negligible. In response, we have removed assertions that the lack of nitrate aerosol is unlikely to cause a systematic bias and that nitrate is "standard in some regional air quality models".

**Similarly, the fact that the authors are comparing these scenarios does not seem to support the argument regarding WACCM's known dust bias that "[b]ecause the same definition of PM2.5 is applied consistently across both scenarios, any systematic bias in the representation of dust is expected to cancel out when examining the relative effects of SAI". The problem with this argument is that, if dust exposure is overestimated by (say) 50%, then so too is the absolute change in dust exposure from which mortality estimates are calculated. This calls into question conclusions about dust being a primary driver (e.g. line 353), as well as figures such as Figure 2 which show dust as being the dominant contributor to changes in PM2.5 in many locations. I would recommend at least considering how the conclusions would be affected if the change in dust is**

**Response to Reviewer #1**

**overestimated relative to changes in other constituents, perhaps using work such as Hancock's to determine the likely overestimate in dust concentrations (and therefore in estimates of absolute changes in dust between scenarios). If anything this supports the authors' conclusions that changes in PM2.5 due to SAI are expected to be small.**

We agree that the previous wording may have overstated the extent to which dust-related biases might cancel when comparing ARISE-SAI-1.5 and SSP2-4.5. As the reviewer notes, if dust concentrations are systematically overestimated, then the absolute magnitude of changes in dust (and thus dust-attributable PM2.5 changes and mortality) may also be overestimated. This uncertainty has implications for interpreting Fig. 2 and for statements identifying dust as a primary driver of PM2.5 differences. In response, we have revised the manuscript to explicitly acknowledge, in the discussion accompanying Fig. 2, that WACCM's dust overestimation could inflate both absolute dust burdens and their apparent contribution to PM2.5 changes. In particular, we clarify that dust's prominence in Fig. 2 should therefore be interpreted with caution, and that dust may appear as a dominant PM2.5 constituent in part due to this known model bias. We also emphasize (as the reviewer points out) that this uncertainty reinforces, rather than contradicts, our broader conclusion: namely, that the air-quality impacts of SAI are small compared with projected policy-driven improvements, and that the absolute contribution of sulfate is modest in the context of future air-quality change.

**On a more minor note, the authors state with regards to the dust bias that they "have included text in the manuscript to highlight this caveat". I cannot find any such additional text in the manuscript, so I would recommend that the authors make clear this caveat.**

We thank the reviewer for pointing this out. We made sure this time the caveat is included in the discussion of the model limitations under Sec. 2.1.

**Finally, I realised that the title might be confusing to some readers. As it stands, it reads "Air quality impacts of stratospheric aerosol injections are likely small and mainly driven by changes in climate, not deposition". However lines 413-416 state that "regional changes in PM2.5 concentrations and the corresponding health impacts are mainly driven by shifts in precipitation patterns and/or circulation, which affect the wet removal of non-sulfate species such as dust and secondary organic aerosols". I fear that the term "deposition" in the title is liable to lead to confusion; I assume the authors mean "settling of injected stratospheric aerosol to the surface", but an air quality expert may instead read it as "wet deposition (i.e. precipitation-related scavenging)". I would recommend modifying the title to make clear exactly what form of deposition is intended, e.g. "...changes in climate, not descent of stratospheric aerosol to the surface" (or ideally something less wordy).**

We have changed the title of this manuscript to "Air quality impacts of stratospheric aerosol injections are likely small and mainly driven by changes in climate, not aerosol settling" to address the reviewer's comments.

**Response to Reviewer #2**

Reviewer comments are in **bold** and the authors' responses are in blue.

We thank the reviewer for their thorough and constructive feedback, which has improved the clarity of the manuscript. In response, we clarified ambiguous language, removed redundant definitions, standardized terminology, and strengthened our discussion of ensemble robustness.

**Introduction: Line 28: "release of precursors such as sulfur dioxide" – precursors to what? Admittedly, readers are not likely to be confused by this, but the sentence should be made precise.**

We thank the reviewer for pointing out this ambiguity. We have revised the sentence to clarify that sulfur dioxide (SO2) is the precursor to sulfate aerosols.

**Model description: Line 93: CESM2(WACCM6) is already defined (and used) as an abbreviation earlier in the text (lines 39-40, albeit without version).**

We have removed the repeated definition in the Model Description section and now simply refer to CESM2(WACCM6) using the abbreviation established earlier in the Introduction.

**Line 109: Suggest moving "(Wen et al, 2023; Wei and Tahrin, 2024)" to end of sentence in line 110, considering this second part contains the findings of said papers.**

The citations have been moved to the end of the sentence.

**Line 131: "with the aim of maintain[ing]"**

The text has been revised accordingly.

**Results: Prior to "In Fig. 1, …" I would highly recommend the authors to add a topic sentence, with the main conclusion of this paragraph. This is completely optional, but paragraph 3.1 currently reads like a summation of figures first and foremost, and less like an active interpretation of these findings. A topic sentence with the main findings before diving into the figures would be a welcome addition here.**

To improve the clarity and flow of Section 3, we have moved the original first paragraph of Section 3.1 out of the subsection and placed it at the beginning of the Results section. This paragraph provides an overarching explanation of the three-way comparison used throughout the analysis and therefore serves more appropriately as general guidance for how the results should be interpreted, rather than as an introductory sentence to Section 3.1.

Because this paragraph establishes the interpretive framework for the analysis, we do not believe it would be appropriate to summarize the findings of Section 3.1 & 3.2 before presenting the underlying diagnostics, mechanisms, and figures that support those findings.

**Response to Reviewer #2**

**The results section often mentions ensemble spread but does not clearly identify which results are robust across the ensemble and which are dominated by variability. For example, the hemispheric ozone asymmetry appears as a strong and consistent feature, whereas several PM2.5 patterns do not. Clarifying this distinction will help readers assess confidence in each result.**

**Related to this, many maps contain extensive stippled regions indicating non-significant differences. In some cases, the text still describes spatial patterns along these regions. It would be helpful to explicitly state when results are not statistically significant rather than implying interpretability from noisy patterns.**

In the revised manuscript, we now explicitly distinguish between ensemble-robust features and those that exhibit substantial internal variability. Specifically:

- We emphasize that the hemispheric ozone asymmetry is a strong, statistically significant, and consistent feature across all ensemble members.
- For PM2.5, we now state clearly that many spatial patterns show large ensemble spread and are therefore less robust, with only a few regions exhibiting statistically significant or ensemble-consistent changes.
- We also updated the description of Fig. 2 to note how stippled areas reflect limited ensemble agreement, but do not affect the overall conclusions that can be made that non-sulfate species dominate PM2.5.
- We have revised the text to explicitly note that many of the PM2.5-related mortality changes occur in regions where internal variability dominates, consistent with the broader PM2.5 spatial pattern being statistically insignificant across much of the globe.

**Line 216: "ITCZ" is defined, but not used again, so this (ITCZ) can be omitted.**

"ITCZ" has been removed since it is not used again.

**Figure 3: I suggest being consistent and writing out "percent" all the time, or even use "relative" instead of switching between "percent" and "%".**

We have revised the figure caption and all associated text to use consistent terminology. Specifically, we now write out "%" throughout the manuscript for clarity and uniformity.